# CONTROLLABLE PARETO MULTI-TASK LEARNING

## ABSTRACT

A multi-task learning (MTL) system aims at solving multiple related tasks at the same time. With a fixed model capacity, the tasks would be conflicted with each other, and the system usually has to make a trade-off among learning all of them together. Multiple models with different preferences over tasks have to be trained and stored for many real-world applications where the trade-off has to be made online. This work proposes a novel controllable Pareto multi-task learning framework, to enable the system to make real-time trade-off switch among different tasks with a single model. To be specific, we formulate the MTL as a preference-conditioned multiobjective optimization problem, for which there is a parametric mapping from the preferences to the Pareto stationary solutions. A single hypernetwork-based multi-task neural network is built to learn all tasks with different trade-off preferences among them, where the hypernetwork generates the model parameters conditioned on the preference. For inference, MTL practitioners can easily control the model performance based on different trade-off preferences in real-time. Experiments on different applications demonstrate that the proposed model is efficient for solving various multi-task learning problems.

## 1 INTRODUCTION

Multi-task learning (MTL) is important for many real-world applications, such as computer vision (Kokkinos, 2017), natural language processing (Subramanian et al., 2018), and reinforcement learning (Van Moffaert & Nowé, 2014). In these problems, multiple tasks are needed to be learned at the same time. An MTL system usually builds a single model to learn several related tasks together, in which the positive knowledge transfer could improve the performance for each task. In addition, using one model to conduct multiple tasks is also good for saving storage costs and reducing the inference time, which could be crucial for many applications (Standley et al., 2020).

However, with fixed learning capacity, different tasks could be conflicted with each other, and can not be optimized simultaneously (Zamir et al., 2018). The practitioners might need to carefully design and train the tasks into different groups to achieve the best performance (Standley et al., 2020). A considerable effort is also needed to find a set of suitable weights to balance the performance of each task (Kendall et al., 2018; Chen et al., 2018b; Sener & Koltun, 2018). For some applications, it might need to train and store multiple models for different trade-off preferences among the tasks (Lin et al., 2019).

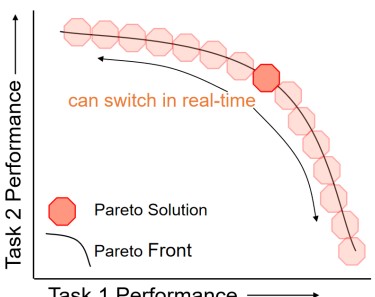

Figure 1: **Controllable Pareto MTL** allows practitioners to control the trade-offs among tasks in real time with a single model, which could be desirable for many MTL applications.

In many real-world MTL applications, the system will need to make a trade-off among different tasks in real time, and it is desirable to have the whole set of optimal trade-off solutions. For example, in a self-driving system, multiple tasks must be conducted simultaneously but also compete for a fixed resource (e.g., fixed total inference time threshold), and their preferences could change in real time for different scenarios (Karpathy, 2019). A recommendation system needs to balance multiple criteria among different stakeholders simultaneously, and making trade-off adjustment would be a crucial component (Milojkovic et al., 2019). Consider the huge storage cost, it is far from ideal to

train and store multiple models to cover different trade-off preferences, which is also not good for real-time adjustment.

In this paper, we propose a novel controllable Pareto multi-task learning framework, to learn the whole trade-off curve for all tasks with a single model. As shown in Fig. 1, MTL practitioners can easily control the trade-off among tasks based on their preferences. To our best knowledge, this is the first approach to learn the MTL trade-off curve. The main contributions are:

- We formulate solving an MTL problem as a preference-conditioned multiobjective optimization problem, and propose a novel Pareto solution generator to learn the whole trade-off curve of Pareto stationary solutions. The proposed Pareto solution generator is also a novel contribution to multiobjective optimization.

- We propose a general hypernetwork-based multi-task neural network framework, and develop an end-to-end optimization algorithm to train a single model concerning different trade-off preferences among all tasks simultaneously. With the proposed model, MTL practitioners can control the trade-offs among different tasks in real time.

## 2 RELATED WORK

**Multi-Task Learning.** The current works on deep multi-task learning mainly focus on designing novel network architecture and constructing efficient shared representation among tasks (Zhang & Yang, 2017; Ruder, 2017). Different deep MTL networks, with hard or soft parameters sharing structures, haven been proposed in the past few years (Misra et al., 2016; Long et al., 2017; Yang & Hospedales, 2017). However, how to properly combine and learn different tasks together remains a basic but challenging problem for MTL applications. Although it has been proposed for more than two decades, the simple linear tasks scalarization approach is still the current default practice to combine and train different tasks in MTL problems (Caruana, 1997).

Some adaptive weight methods have been proposed to better combine all tasks in MTL problems with a single model (Kendall et al., 2018; Chen et al., 2018b; Liu et al., 2019; Yu et al., 2020). However, analysis on the relations among tasks in transfer learning (Zamir et al., 2018) and multi-task learning (Standley et al., 2020) show that some tasks might conflict with each other and can not be optimized at the same time. Sener and Koltun (Sener & Koltun, 2018) propose to treat MTL as a multiobjective optimization problem, and find a single Pareto stationary solution among different tasks. Pareto MTL (Lin et al., 2019) generalizes this idea, and proposes to generate a set of Pareto stationary solutions with different trade-off preferences. Recent works focuses on generating diverse and dense Pareto stationary solutions (Mahapatra & Rajan, 2020; Ma et al., 2020).

**Multiobjective Optimization.** Multiobjective optimization itself is a popular research topic in the optimization community. Many gradient-based and gradient-free algorithms have been proposed in the past decades (Fliege & Svaiter, 2000; Désidéri, 2012; Miettinen, 2012). In addition to MTL, they also can be used in reinforcement learning (Van Moffaert & Nowé, 2014) and neural architecture search (NAS) (Elsken et al., 2019). However, most methods directly use or modify well-studied multiobjective algorithms to find a single solution or a finite number of Pareto stationary solutions.

Recently, Lu et al. (2020) proposed a novel supernet-based multi-objective neural architecture transfer framework. This method simultaneously optimizes a set of neural networks, which are sampled from a large supernet, to approximate the optimal trade-off curve among different objectives via a single run. The obtained supernet also supports fast adaption to new tasks and domains. Some efforts have been made to learn the entire trade-off curve for different machine learning applications. Parisi et al. (2016) proposed to learn the Pareto manifold for a multi-objective reinforcement learning problem. However, since this method does not consider the preference for generation, it does not support real-time preference-based adjustment. Very recently, some efforts have been made to learn preference-based solution adjustment for multi-objective reinforcement learning (Yang et al., 2019) and image generation (Dosovitskiy & Djolonga, 2020) with simple linear combinations. There is a concurrent work (Anonymous, 2021) that also proposes to learn the entire trade-off curve for MTL problems by hypernetwork. While this work emphasizes the runtime efficiency on training for multiple preferences, we highlight the benefit of scalability and real-time preference adjustment.

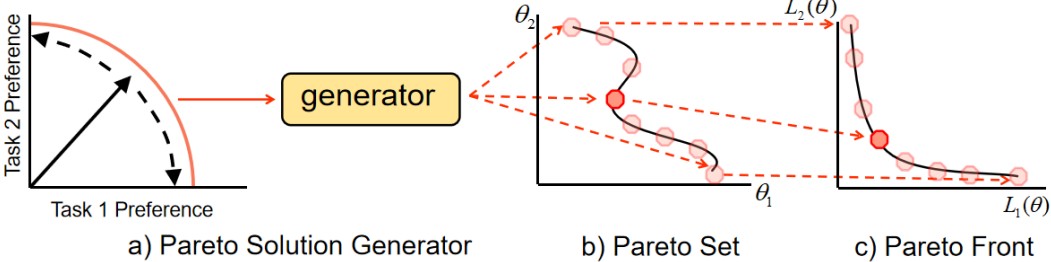

a) Pareto Solution Generator     b) Pareto Set     c) Pareto Front

Figure 2: **Controllable Pareto MTL** can directly generate a corresponding Pareto stationary solution based on a given preference among tasks. (a) A practitioner adjusts the $m$-dimensional preference vector $\boldsymbol{p}$ to specify a trade-off preference among $m$ tasks. (b) A trained Pareto solution generator directly generates a Pareto solution $\boldsymbol{\theta_p}$ conditioned on $\boldsymbol{p}$. (c) The generated solution $\boldsymbol{\theta_p}$ should have objective values $L(\boldsymbol{\theta_p})$ that satisfies the trade-off preference among $m$ tasks. The number of possible preference vectors and the corresponding Pareto stationary solutions could be infinite for an MTL problem. We develop the idea in Section 4 and propose the MTL model in Section 5.

**HyperNetworks.** The hypernetwork approach is initially proposed for dynamic modeling and model compression (Schmidhuber, 1992; Ha et al., 2017). It also leads to various novel applications such as hyperparameter optimization (Brock et al., 2018; MacKay et al., 2019), Bayesian inference (Krueger et al., 2018; Dwaracherla et al., 2020), and transfer learning (von Oswald et al., 2020; Meyerson & Miikkulainen, 2019). Recently, some discussions have been made on its initialization method (Chang et al., 2020), the optimization dynamic (Littwin et al., 2020), and its relation to other multiplicative interaction methods (Jayakumar et al., 2020). This paper uses a hypernetwork to generate the weights of the main multi-task neural network conditioned on different preferences.

## 3 MTL AS MULTI-OBJECTIVE OPTIMIZATION

An MTL problem involves learning multiple related tasks at the same time. For training a deep multi-task neural network, it is also equal to minimize the losses for multiple tasks:

$$\min_\theta \mathcal{L}(\theta) = (\mathcal{L}_1(\theta), \mathcal{L}_2(\theta), \cdots, \mathcal{L}_m(\theta)), \tag{1}$$

where $\theta$ is the neural network parameters and $\mathcal{L}_i(\theta)$ is the empirical loss of the $i$-th task. For learning all $m$ tasks together, an MTL algorithm usually aims to minimize all the losses at the same time. However, in many MTL problems, it is impossible to find a single best solution to optimize all the losses simultaneously. With different trade-offs among the tasks, the problem (1) could have a set of Pareto solutions which satisfy the following definitions (Zitzler & Thiele, 1999):

**Pareto dominance.** Let $\theta_a, \theta_b$ be two solutions for problem (1), $\theta_a$ is said to dominate $\theta_b$ ($\theta_a \prec \theta_b$) if and only if $\mathcal{L}_i(\theta_a) \leq \mathcal{L}_i(\theta_b), \forall i \in \{1, ..., m\}$ and $\mathcal{L}_j(\theta_a) < \mathcal{L}_j(\theta_b), \exists j \in \{1, ..., m\}$.

**Pareto optimality.** $\theta^*$ is a Pareto optimal solution if there does not exist $\hat{\theta}$ such that $\hat{\theta} \prec \theta^*$. The set of all Pareto optimal solutions is called the Pareto set. The image of the Pareto set in the objective space is called the Pareto front.

**Approximation with Finite Pareto Stationary Solutions.** The number of Pareto solutions could be infinite, and their objective values are on the boundary of the valid value region (Boyd & Vandenberghe, 2004). Under mild conditions, the whole Pareto set and Pareto front would be $(m - 1)$ dimensional manifolds in the solution space and objective space, respectively (Miettinen, 2012). In addition, for a general multiobjective optimization problem, no method can guarantee to find a Pareto optimal solution. Traditional multiobjective optimization algorithms aim at finding a set of finite Pareto stationary solutions to approximate the whole Pareto set and Pareto front:

$$\hat{S} = \{\hat{\boldsymbol{\theta}}_1, \hat{\boldsymbol{\theta}}_2, \cdots, \hat{\boldsymbol{\theta}}_K\}, \qquad \hat{F} = \{\mathcal{L}(\hat{\boldsymbol{\theta}}_1), \mathcal{L}(\hat{\boldsymbol{\theta}}_2), \cdots, \mathcal{L}(\hat{\boldsymbol{\theta}}_K)\}, \tag{2}$$

where $\hat{S}$ is the approximated Pareto set of $K$ estimated solutions, and $\hat{F}$ is the set of corresponding objective vectors in the objective space. The current multiobjective optimization based MTL methods aim to find a single ($K = 1$) (Sener & Koltun, 2018) or a set of multiple ($K > 1$) Pareto stationary solutions (Lin et al., 2019; Mahapatra & Rajan, 2020; Ma et al., 2020) for a given problem.

## 4 PREFERENCE-CONDITIONED PARETO SOLUTION GENERATOR

Instead of finite Pareto set estimation, we propose to directly learn the whole trade-off curve for a MTL problem in this paper. As shown in Fig. 2, we want to build a Pareto solution generator to map a preference vector $\boldsymbol{p}$ to its corresponding Pareto stationary solution $\boldsymbol{\theta_p}$. If an optimal generator $\boldsymbol{\theta_p} = g(\boldsymbol{p}|\boldsymbol{\phi}^*)$ is obtained, MTL practitioners can assign their preference via the preference vector $\boldsymbol{p}$, and directly obtain the corresponding solution $\boldsymbol{\theta_p}$ with the specific trade-off among tasks.

With the Pareto solution generator, we can obtain the approximate Pareto set/front as:

$$\hat{S} = \{\boldsymbol{\theta_p} = g(\boldsymbol{p}|\boldsymbol{\phi}^*)|\boldsymbol{p} \in \boldsymbol{P}\}, \qquad \hat{F} = \{\mathcal{L}(\boldsymbol{\theta_p})|\boldsymbol{\theta_p} \in \hat{S}\}, \tag{3}$$

where $\boldsymbol{P}$ is the set of all valid preference vectors, $g(\boldsymbol{p}|\boldsymbol{\phi}^*)$ is the Pareto generator with optimal parameters $\phi*$. Once we have a proper generator $g(\boldsymbol{p}|\boldsymbol{\phi}^*)$, we can reconstruct the whole approximated set of Pareto stationary solutions $\hat{S}$ and the corresponding trade-off curve $\hat{F}$ by going through all possible preference vector $\boldsymbol{p}$ from $\boldsymbol{P}$. In the rest of this section, we discuss two approaches to define the form of preference vector $\boldsymbol{p} \in \boldsymbol{P}$ and its connection to the corresponding solution $\boldsymbol{\theta}_p = g(\boldsymbol{p}|\boldsymbol{\phi}^*)$.

**Preference-Conditioned Linear Scalarization:** A simple and straightforward approach is to define the preference vector $\boldsymbol{p}$ and the corresponding solution $\boldsymbol{\theta_p}$ via the weighted linear scalarization:

$$\boldsymbol{\theta_p} = g(\boldsymbol{p}|\boldsymbol{\phi}^*) = \arg\min_{\boldsymbol{\theta}} \sum_{i=1}^{m} \boldsymbol{p}_i \mathcal{L}_i(\boldsymbol{\theta}), \tag{4}$$

where the preference vector $\boldsymbol{p} = (\boldsymbol{p}_1, \boldsymbol{p}_2, \cdots, \boldsymbol{p}_m)$ is the weight for each task, and the corresponding solution $\boldsymbol{\theta_p}$ is the minimum of the weighted linear scalarization. If we further require $\sum \boldsymbol{p}_i = 1$, the set of all valid preference vector $\boldsymbol{P}$ is an $(m-1)$-dimensional manifold in $\mathbb{R}^m$. Therefore, the valid preference set and the Pareto set are both $(m-1)$-dimensional manifolds.

Although this approach is straightforward, it is not optimal for multiobjective optimization. Linear scalarization can not find any Pareto solution on the non-convex part of the Pareto front (Das & Dennis, 1997; Boyd & Vandenberghe, 2004). In other words, unless the problem has a convex Pareto front, the generator defined by linear scalarization cannot cover the whole Pareto set manifold.

**Preference-Conditioned Multi-Objective Optimization:** To better approximate the manifold of Pareto set, we generalize the idea of decomposition-based multiobjective optimization (Zhang & Li, 2007; Liu et al., 2014) and Pareto MTL (Lin et al., 2019) to connect the preference vector and the corresponding Pareto solution. To be specific, we define the preference $\boldsymbol{p}$ as an $m$ dimensional unit vector in the loss space, and the corresponding solution $\boldsymbol{\theta_p}$ is the one on the Pareto front which has the smallest angle with $\boldsymbol{p}$.

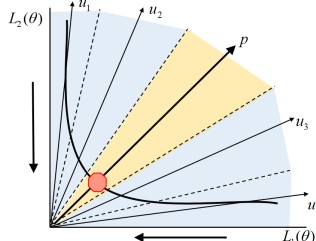

Figure 3: An MTL problem is decomposed by a set of unit vectors in the loss space.

The idea is illustrated in in Fig. 3. With a set of randomly generated unit reference vectors $\boldsymbol{U} = \{\boldsymbol{u}^{(1)}, \cdots, \boldsymbol{u}^{(K)}\}$ and the preference vector $\boldsymbol{p}$, an MTL problem is decomposed into different regions in the loss space. We call the region closest to $\boldsymbol{p}$ as its preferred region. The corresponding Pareto solution $\boldsymbol{\theta_p}$ is the solution that always belongs to the preferred region and on the Pareto front. Formally, we can define the corresponding Pareto solution as:

$$\boldsymbol{\theta_p} = g(\boldsymbol{p}|\boldsymbol{\phi}^*) = \arg\min_{\boldsymbol{\theta}} \mathcal{L}(\boldsymbol{\theta}) \text{ s.t. } \mathcal{L}(\boldsymbol{\theta}) \in \Omega(\boldsymbol{p}, \boldsymbol{U}), \tag{5}$$

where $\mathcal{L}(\boldsymbol{\theta})$ is the loss vector and $\Omega(\boldsymbol{p}, \boldsymbol{U})$ is the constrained preference region conditioned on the preference and reference vectors $\Omega(\boldsymbol{p}, \boldsymbol{U}) = \{\boldsymbol{v} \in R_+^m | \angle(\boldsymbol{v}, \boldsymbol{p}) \leq \angle(\boldsymbol{v}, \boldsymbol{u}^{(j)}), \forall j = 1, ..., K\}$.

The constraint is satisfied if the loss vector $\mathcal{L}(\boldsymbol{\theta})$ has the smallest angle with the preference vector $\boldsymbol{p}$. The corresponding solution $\boldsymbol{\theta_p}$ is a restricted Pareto optimal in $\Omega(\boldsymbol{p}, \boldsymbol{U})$. Since we require the preference vectors should be unit vector $||\boldsymbol{p}||^2 = 1$ in the $m$-dimensional space, the set of all valid preference vectors $\boldsymbol{P}$ is an $(m-1)$ manifold in $\mathbb{R}^m$. Similar to the linear scalarization case, a preference vector might have more than one corresponding solution, and a Pareto solution can correspond to more than one preference vector. For simplicity, we assume there is a one-to-one mapping between the preference vector and the corresponding Pareto solution in this paper.

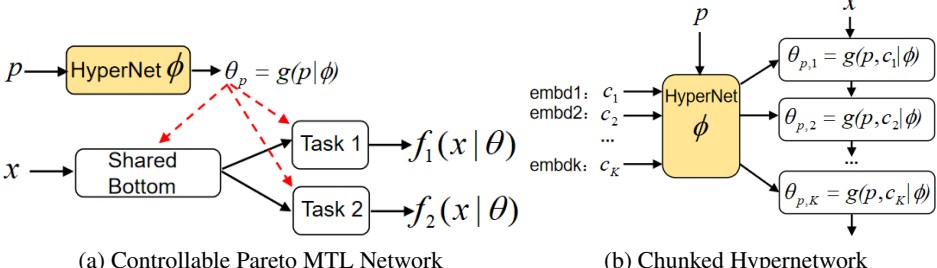

**Figure 4: The Controllable Pareto Multi-Task Network:** (a) The main MTL network has a fixed structure, and all its parameters are generated by the hypernetwork conditioned on the preference vector. (b) With the chunking method, the hypernetwork can iteratively generate parameters for different parts of the main network.

## 5 CONTROLLABLE PARETO MULTI-TASK LEARNING

In this section, we propose a hypernetwork-based deep multi-task neural network framework, along with an efficient end-to-end optimization procedure to solve the MTL problem.

### 5.1 HYPERNETWORK-BASED DEEP MULTI-TASK NETWORK

The proposed controllable Pareto multi-task network is shown in Fig. 4(a). As discussed in the previous section, we use a preference vector to represent a practitioner's trade-off preference among different tasks. The hypernetwork takes the preference vector $p$ as its input, and generates $\theta_p = g(p|\phi)$ as the corresponding parameters for the main MTL network. the only trainable parameters to be optimized are the hypernetwork parameters $\phi$. Practitioners can easily control the MTL network's parameters, and hence the performances on different tasks in real time, by simply adjusting the preference vector $p$.

**Model Structure and Regularization:** In our proposed model, the main MTL network always has a fixed structure, and the parameters $\theta_p$ are generated by the hypernetwork. We consider the main MTL network with hard-shared encoder (Ruder, 2017; Vandenhende et al., 2020) as shown in Fig. 4(a). Once the parameters $\theta_p$ are generated, an input $x$ to the main MTL network will first go through the shared encoder, and then all task-specific heads to obtain the outputs $f_i(x|\theta_p)$. Different tasks share the same encoder, and they are regularized by each other as in the traditional MTL model. Our proposed model puts the cross-tasks regularization on the hypernetwork, where it should generate a set of good encoder parameters that work well for all tasks with a given preference.

There is also a regularization effect across different trade-off preferences. Our goal is to train a single hypernetwork-based model that performs well for all preferences. In other words, the generated main MTL network with a specific preference is regularized by the other networks with different preferences through the hypernetwork. This regularization effect might be best explained by the batched preferences update discussed in Appendix B.3. In this respect, it is similar to von Oswald et al. (2020) that uses hypernetwork with explicit regularization on separate tasks in the continual learning setting. It is also good for knowledge transfer among different preferences. We further discuss how to use our model with a pretrained feature extractor in Appendix A.

**Chunked Hypernetwork:** With the model compression ability powered by the hypernetwork, our proposed model can scale well to large scale problems. Chunking (Ha et al., 2017; von Oswald et al., 2020) is a commonly used method to reduce the number of parameters for the hypernetwork. As shown in Fig. 4(b), a hypernetwork can iteratively generate small parts of the main network $\theta_p = [\theta_{p,1}, \theta_{p,2}, \cdots, \theta_{p,K}]$, with a reasonable model size and multiple trainable chunk embedding $\{c_k\}_{k=1}^K$, where $\theta_{p,k} = g(p, c_k|\phi)$. In this way, we can significantly compress the model size and make it scalable for large MTL models. In this paper, we use fully-connected networks as the hypernetwork, and the main MTL neural networks have the same structures as the models used in current MTL literature (Liu et al., 2019; Sener & Koltun, 2018; Vandenhende et al., 2020).

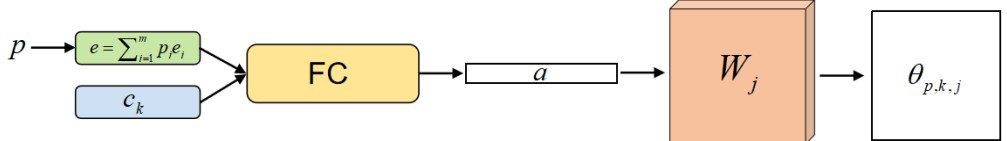

Figure 5: **Parameter Generation:** The hypernetwork takes a preference vector $p$ as input and generates the parameter $\theta_{p,k,j}$ for part of the main MTL network, where $e = \sum_{i=1}^{m} p_i e_i$ is the preference embedding, $c_k$ is the chunk embedding, **FC** is the fully connected layers, $a$ is the generated vector, and $W_j$ is a parameter tensor. All blocks in color contain trainable parameters for the hypernetwork.

**Parameter Generation:** We follow the hypernetwork design proposed by Ha et al. (2017) with additional chunk embedding (von Oswald et al., 2020) to construct our hypernetwork-based model. An illustration on the process to generate a chunk of parameters for the main MTL network is shown in Fig. 5. We first assign $m$ trainable $q$-dimensional embedding $e_i$ for each task, and use the preference-weighted embedding $e = \sum_{i}^{m} p_i e_i$ as the input to the hypernetwork. In this way, the input dimension increases from a small $m$ (e.g., 2) to a larger $q$ (e.g. 100). We simply concatenate the preference embedding $e$ with a chunk embedding $c_k$ as the input to a set of fully connected layers, and obtain the generated vector $a \in R^d$. Then the hypernetwork uses a linear operation to project the vector $a$ into the generated parameters $\theta_{p,k,j} = W_j a$, where $W_j \in R^{n_{j,1} \times n_{j,2} \times d}$ contains $n_{j,1} \times n_{j,2} \times d$ trainable parameters and $\theta_{p,k,j} \in R^{n_{j,1} \times n_{j,2}}$. The hypernetwork generates all parameters $\theta_p = [\theta_{p,1}, \theta_{p,2}, \cdots, \theta_{p,K}]$ by iteratively conducting this generation process.

**Scalablity and Inference:** In the proposed hypernetwork, the fully connected layers are reusable to generate different $a$ based on different chunk embedding. Most hypernetwork parameters are in the parameter tensors. By sharing the same $W_j$ to generate different chunks of parameters with different $a$, the number of parameters for the hypernetwork can be further reduced (Ha et al., 2017), which makes it suitable to generate the parameters for large scale models. In this work, we keep our hypernetwork-based model to have a comparable size with the corresponding MTL model. For inference, the fully connected layers can make batch inference for multiple chunk embedding to generate different $a$, and most computation is on the linear projection. It leads to an acceptable inference latency overhead, and supports real-time trade-off preference adjustment.

## 5.2 END-TO-END OPTIMIZATION: LEARNING PARETO GENERATOR VIA SOLVING MTL

Since we want to control the trade-off preference at the inference time, we need to train a model suitable for all preference vector $p$ in the valid set $P$ rather than a single preference. Suppose we have a probability distribution on all possible preference vector $p \sim P_p$, a general goal would be $\min_{\phi} \mathbb{E}_{p \sim P_p} \mathcal{L}(g(p|\phi))$. It is hard to optimize the parameters within the expectation directly. We use Monte Carlo method to sample the preference vector, and use the stochastic gradient descent algorithm to train the deep neural network. At each step, we can sample one preference vector $p$ and optimize the loss function:

$$\min_{\phi} \mathcal{L}(g(p|\phi)), \quad p \sim P_p. \tag{6}$$

At each iteration $t$, if we have a valid gradient direction $d_t$ for the loss $\mathcal{L}(g(p|\phi_t))$, we can simply update the parameters with gradient descent $\phi_{t+1} = \phi_t - \eta d_t$. For the preference-conditioned linear scalarization case as in problem (4), the calculation of $d_t$ is straightforward:

$$d_t = \sum_{i=1}^{m} p_i \nabla_{\phi_t} \mathcal{L}_i(g(p|\phi_t)), \quad p \sim P_p. \tag{7}$$

For the preference-conditioned multiobjective optimization problem (5), a valid descent direction should simultaneously reduce all losses and activated constraints. One valid descent direction can be written as a linear combination of all tasks with dynamic weight $\alpha_i(t)$:

$$d_t = \sum_{i=1}^{m} \alpha_i(t) \nabla_{\phi_t} \mathcal{L}_i(g(p|\phi_t)), \quad p \sim P_p, \quad U \sim P_U, \tag{8}$$

where the coefficients $\alpha_i(t)$ is depended on both the loss functions $\mathcal{L}_i(g(p|\phi_t))$ and the constraints $\mathcal{G}_j(\theta|p, U)$. We give the detailed derivation in the Appendix B.1 due to page limit.

## 5.3 ALGORITHM FRAMEWORK

**Algorithm 1** Controllable Pareto Multi-Task Learning

1: Initialize the hypernetwork parameters $\phi_0$
2: **for** $t = 1$ to $T$ **do**
3:    Sample a preference vector $p \sim P_p$, $U \sim P_U$
4:    Obtain a valid gradient direction $d_t$ from (7) or (8)
5:    Update the parameters $\phi_{t+1} = \phi_t - \eta d_t$
6: **end for**
7: **Output:** The hypernetwork parameters $\phi_T$

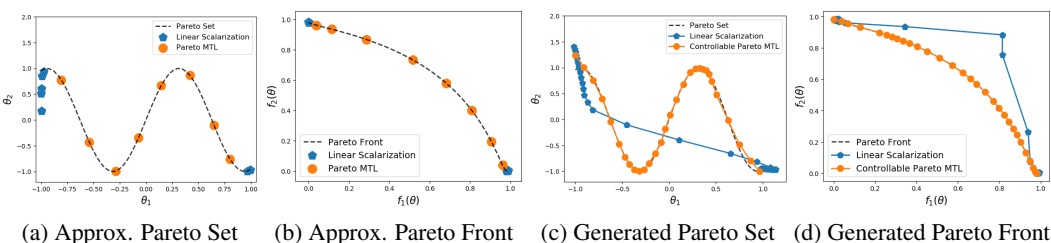

Figure 6: **The algorithm framework and the continually learned trade-off curve for the training loss on MultiMNIST problem:** Our proposed algorithm optimizes the hypernetwork parameters for all valid preferences, and continually learns the trade-off curve during the optimization process.

The algorithm framework is shown in **Algorithm 1**. We use a simple uniform distribution on all valid preferences and references vector in this paper. At each iteration, we calculate a valid gradient direction and update the hypernetwork parameters. The proposed algorithm simultaneously optimizes the hypernetwork for all valid preference vectors that represent different valid trade-offs among all tasks. As shown in Fig. 6, it continually learns the trade-off curve during the optimization process. We obtain a Pareto generator at the end, and can directly generate a solution $\theta_p = g(p|\phi_T)$ from any valid preference vector $p$. By taking all preference vectors as input, we can construct the whole trade-off curve.

## 6 SYNTHETIC MULTI-OBJECTIVE OPTIMIZATION PROBLEM

| (a) Approx. Pareto Set | (b) Approx. Pareto Front | (c) Generated Pareto Set | (d) Generated Pareto Front |

Figure 7: **Point Set Approximation and Generated Pareto Front. (a) & (b):** Traditional multi-objective optimization algorithms can only find a finite point set to approximate the Pareto set and Pareto front. In addition, the simple linear scalarization method can not cover most part of the Pareto set/front when the ground truth Pareto front is a concave curve. **(c) & (d):** Our proposed controllable Pareto MTL method can successfully generate the whole Pareto front from the preference vectors. In contrast, the linear scalarization method is only accurate for Pareto solutions near the endpoints.

In the previous section, we propose to use a hypernetwork to generate the parameters for the main deep multi-task network based on a preference vector. The network parameters are in a high-dimensional decision space, and the optimization landscape would be complicated. To better analyze the behavior and convergence performance of the proposed algorithm, we first use it to generate the Pareto solutions for a low-dimensional multiobjective optimization problem.

The experimental results are shown in Fig. 7. This synthetic problem has a sin-curve-like Pareto set in the solution space, and its corresponding Pareto front is a concave curve in the objective space. The detailed definition is given in the appendix. Our proposed controllable Pareto MTL method can cover and reconstruct the whole manifold of Pareto front from the preference vectors, while the traditional methods can only find a finite point set approximation. It should be noticed that the simple linear scalarization method has poor performance for the problem with concave Pareto front.

# 7 EXPERIMENTS

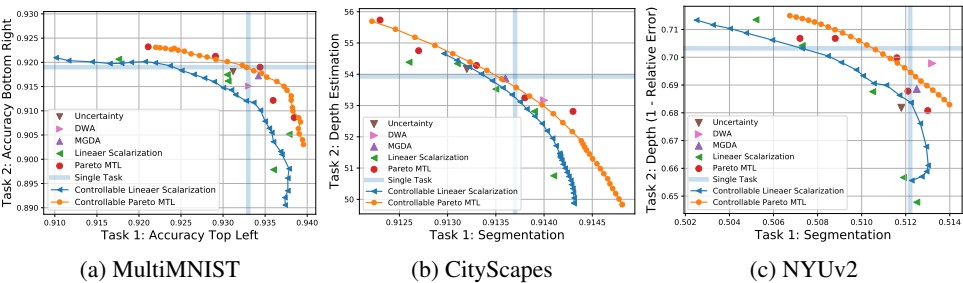

(a) MultiMNIST        (b) CityScapes        (c) NYUv2

Figure 8: **Results on MultiMNIST, CityScapes and NYUv2:** Our proposed algorithm can generate the Pareto front for each problem with a single model respectively. For CityScapes and NYUv2, we report pixel accuracy for segmentation and (1 - relative error) for depth estimation.

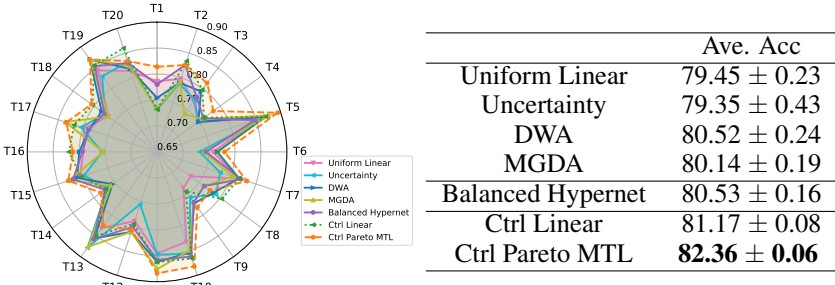

|  | Ave. Acc |
|---|---|
| Uniform Linear | $79.45 \pm 0.23$ |
| Uncertainty | $79.35 \pm 0.43$ |
| DWA | $80.52 \pm 0.24$ |
| MGDA | $80.14 \pm 0.19$ |
| Balanced Hypernet | $80.53 \pm 0.16$ |
| Ctrl Linear | $81.17 \pm 0.08$ |
| Ctrl Pareto MTL | $\mathbf{82.36 \pm 0.06}$ |

Figure 9: **The results of CIFAR-100 with 20 Tasks:** The results for each baseline method is a single solution. Our proposed algorithms switch to the corresponding preference when making prediction for each task (e.g., preference on task 1 for making prediction on task 1).

In this section, we validate the performance of the proposed controllable Pareto MTL to generate the trade-off curves for different MTL problems. We compare it with the following MTL algorithms: **1) Linear Scalarization:** simple linear combination of different tasks with fixed weights; **2)Uncertainty** (Kendall et al., 2018): adaptive weight assignments with balanced uncertainty; **3)DWA** (Liu et al., 2019): dynamic weight average for the losses; **4) MGDA** (Sener & Koltun, 2018): finds one Pareto stationary solution; **5) Pareto MTL** (Lin et al., 2019): generates a set of wildly distributed Pareto stationary solutions; and **6) Single Task:** the single task baseline.

We conduct experiments on the following widely-used MTL problems: **1) MultiMNIST (Sabour et al., 2017):** This problem is to classify two digits on one image simultaneously. **2) CityScapes (Cordts et al., 2016):** This dataset has street-view RGB images, and involves two tasks to be solved, which are pixel-wise semantic segmentation and depth estimation. **3) NYUv2 (Silberman et al., 2012):** This dataset is for indoor scene understanding with two tasks: a 13-class semantic segmentation and indoor depth estimation. **4) CIFAR-100 with 20 Tasks:** Follow a similar setting in (Rosenbaum et al., 2018), we split the original CIFAR-100 dataset (Krizhevsky & Hinton, 2009) into 20 five-class classification tasks. Details for these problems can be found in the Appendix C.

**Result Analysis:** The experimental results are shown in Fig. 8 where all models are trained from scratch. In all problems, our proposed algorithm can learn the trade-off curve with a single model, while the other methods need to train multiple models and cannot cover the entire curve. In addition, the proposed preference-conditioned multiobjective optimization method can always find a better trade-off curve that dominates the curve found by the simple linear scalarization. These results validate the effectiveness of the proposed model and the end-to-end optimization method.

For all experiments, the single-task models are a strong baseline in performance with larger model size ($m$ full models). However, it cannot dominate most of the approximated Pareto front learned by

Table 1: Overveiw of the models we used in different MTL problems.

| Problem | Tasks | Network | Model | Params | Latency(ms) |
|---------|-------|---------|-------|--------|-------------|
| MultiMNIST | 2 | LeNet | Single MTL Model | 84K | $1.49 \pm 0.12$ |
| | | | Hypernetwork-based Model | 83K | $1.96 \pm 0.06$ |
| CityScape | 2 | SegNet | Single MTL Model | 25M | $65.0 \pm 1.23$ |
| | | | Hypernetwork-based Model | 26M | $88.6 \pm 4.85$ |
| NYUv2 | 2 | SegNet | Single MTL Model | 25M | $179 \pm 13.1$ |
| | | | Hypernetwork-based Model | 26M | $198 \pm 11.5$ |
| CIFAR100 | 20 | ConvNet | Single MTL Model | 1.4M | $9.95 \pm 0.17$ |
| | | | Hypernetwork-based Model | 5.9M | $23.4 \pm 0.34$ |
| NYUv2 | 2 | ResNet-50 | Single MTL Model | 56M | $236 \pm 11.2$ |
| | | | Hypernetwork-based Model | 120M | $384 \pm 18.0$ |

our model. Our models provide diverse optimal trade-offs among tasks (e.g., in the upper left and lower right area) for different problems and support real-time adjustment among them.

The experimental result on the 20-tasks CIFAR100 problem is shown in Fig. 9. Our proposed model can achieve the best overall performance among different tasks by making a real-time trade-off adjustment. It also outperforms the balanced hypernet approach, which uses the same hypernetwork-based model to optimize all tasks simultaneously. This result shows the benefit of our proposed model on preference-based modeling and real-time preference adjustment.

**Scalability and Latency:** The models we use in the experiments are summarized in Table.1. As discussed in the previous sections, we build the hypernetwork such that the proposed models have a comparable number of parameters with the corresponding MTL model. Given the current works (Lin et al., 2019; Mahapatra & Rajan, 2020; Ma et al., 2020) need to train and store multiple MTL models to approximate the Pareto front, our proposed model is much more parameter-efficient while learning the whole trade-off curve. In this sense, it is more scalable than the current methods.

The inference latency for all models are also reported. The latency is measured on a 1080Ti GPU with the training batch size for each problem. We report the averaged mean and standard deviation over 100 independent runs. For our proposed model, we randomly adjust the preference for each batch. Our model has an affordable overhead on the inference latency, which is suitable for real-time preference adjustment.

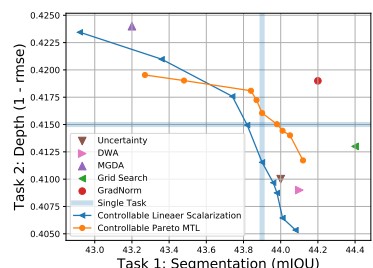

Figure 10: **Results on NYUv2 with ResNet-50 Backbone:** The proposed model can generate reasonable trade-off curves.

**Experiment with Pretrained Backbone:** We conduct an experiment on the NYUv2 problem with a large-scale ResNet-50 backbone (He et al., 2016) following the experimental setting in Vandenhende et al. (2020). Each model has a ResNet-50 backbone pretrained on ImageNet, and task-specific heads with ASPP module (Chen et al., 2018a). The results are shown in Fig.10, where the results for MTL models are from Vandenhende et al. (2020) with the best finetuned configurations. Our proposed model also uses the pretrained ResNet-50 backbone, and generates the encoder's last few layers and all task-specific heads. Our model can learn the trade-off curve with comparable performance, which might be further improved with task balancing methods and better hypernetwork architecture design as in Vandenhende et al. (2020).

## 8 CONCLUSION

In this paper, we proposed a novel controllable Pareto multi-task learning framework for solving MTL problems. With a hypernetwork-based Pareto solution generator, our method can directly learn the whole trade-off curve for all tasks with a single model. It allows practitioners to easily make real-time trade-off adjustment among tasks at the inference time. Experimental results on various MTL applications demonstrated the usefulness and efficiency of the proposed method.

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

APPENDIX

We provide more discussion and analysis in this appendix, which can be summarized as follows:

- **Model with Pretrained Feature Extractor:** We discuss how to use pretrained feature extractor in our proposed model in Appendix A.
- **Preference-based Multiobjective Gradient Descent:** We give the details of preference-based multiobjective gradient descent for model training in Appendix B.
- **Experimental Setting:** The detailed experimental settings are provided in Section C.

## A    MODEL WITH PRETRAINED FEATURE EXTRACTOR

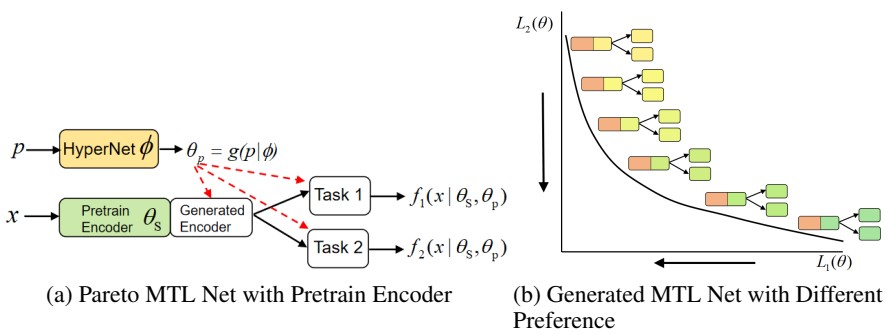

(a) Pareto MTL Net with Pretrain Encoder

(b) Generated MTL Net with Different Preference

Figure 11: (a) Controllable Pareto MTL network with a pretrained feature extractor: The pretrained encoder is preference-agnostic, and the hypernetwork generates the rest parts of the main MTL network. (b) The generated MTL models share the same parameters for part of the encoder.

One powerful and efficient way to improve the MTL model performance is to use a feature extractor pretrained on large scale dataset (Ruder, 2017; Vandenhende et al., 2020). This approach can be easily incorporated into our proposed model. As shown in Fig.11(a), our model can be built on top of a shared pretrained feature extractor, and the hypernetwork generates the parameters for the rest part of the encoder and all task-specific heads. The main MTL model parameters $\theta = [\theta_s, \theta_p]$ now have a preference-agnostic part $\theta_s$ and a preference-conditioned part $\theta_p = g(p|\phi)$.

In other words, no matter what preferences are given, all generated MTL models will share the parameters for the feature extractor as shown in Fig.11(b). It is a structure constraint across all generated solutions, and the trade-off curve learned by our model would be a front of restricted Pareto stationary solutions, with constraints on the sharing parameters. The shared encoder brings extra knowledge from the datasets it pretrained on, and it also has a regularization effect since it is shared by models with different preferences. The experiment on models with ResNet-50 backbone validates that our proposed model can still learn the trade-off curve with a pretrained encoder.

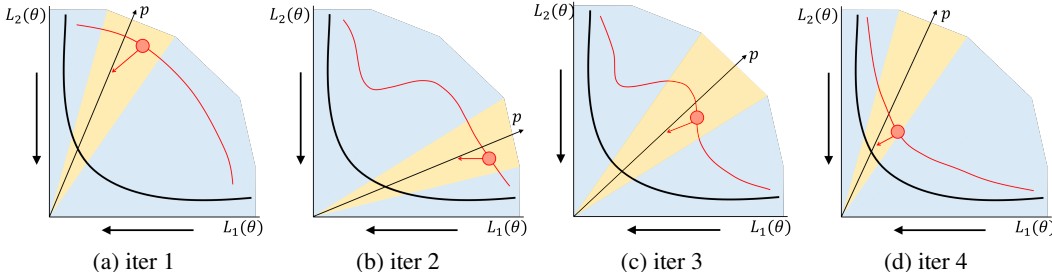

|(a) iter 1|(b) iter 2|(c) iter 3|(d) iter 4|

Figure 12: **The training process of the Pareto solution generator.** At each iteration, the proposed algorithm randomly samples a preference vector and a set of reference vectors to decompose the loss space, and calculates a valid gradient direction to update the Pareto solution generator. The algorithm iteratively learns and improves the generated manifold during the optimization process.

## B PREFERENCE-BASED MULTIOBJECTIVE GRADIENT DESCENT

In this section, we give the details of preference-based multiobjective gradient descent for training the hypernetwork-based model.

### B.1 PREFERENCE-CONDITIONED MULTIOBJECTIVE GRADIENT DESCENT

The Pareto solution generator generates a solution $\boldsymbol{\theta_p} = g(\boldsymbol{p}|\boldsymbol{\phi})$ from the preference vector $\boldsymbol{p}$. Under ideal condition, if we take all valid preference vectors $\boldsymbol{p} \in \boldsymbol{P}$ (which is an $(m-1)$-dimensional manifold) as input, the output would be another $(m-1)$-dimensional manifold. At the end of the algorithm, we hope the generated manifold is close to the manifold of true Pareto set.

In the proposed algorithm, we randomly sample a preference vector $\boldsymbol{p}$ at each iteration to update the generator parameters $\phi$. As illustrated in Fig. 12, it continually learns and improves the current manifold around the preference vector at each iteration. It should be noticed that the sampled preference vector and all reference vectors are different at each iteration, so as the decomposition.

As mentioned in the **Algorithm 1** in the main paper, we use simple gradient descent to update the Pareto solution generator at each iteration. For the case of linear scalarization, calculating the gradient direction is straightforward. In this subsection, we discuss how to obtain a valid gradient direction for the preference-conditioned multiobjective optimization. Similar to the previous work (Sener & Koltun, 2018; Lin et al., 2019), we use multiobjective gradient descent (Fliege & Svaiter, 2000; Désidéri, 2012) to solve the MTL problem. However, in our algorithm, the optimization parameter is $\phi$ for the Pareto solution generator rather than $\theta$ for a single solution.

A simple illustration of the multiobjective gradient descent idea for a problem with two tasks is shown in Fig. 13. At each iteration, the algorithm samples a preference vector $\boldsymbol{p}$ and obtains its current corresponding solution $\boldsymbol{\theta_p}$. A valid gradient direction should reduce all the losses and guide the generated solution $\boldsymbol{\theta_p} = g(\boldsymbol{p}|\boldsymbol{\phi})$ toward the preference region $\Omega(\boldsymbol{p}, \boldsymbol{U})$ around the preference vector $\boldsymbol{p}$.

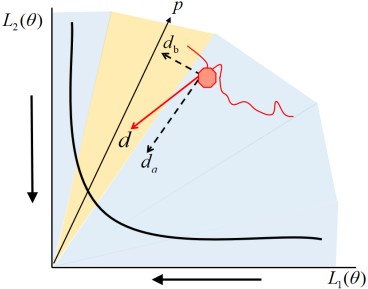

Figure 13: For a preference vector, the valid gradient direction should reduce all losses and activated constraints for the generated solution.

At iteration $t$, for a given preference vector $\boldsymbol{p}$, the preference-conditioned multiobjective problem with the Pareto solution generator can be written as:

$$\min_{\boldsymbol{\phi}_t}(\mathcal{L}_1(g(\boldsymbol{p}|\boldsymbol{\phi}_t)), \mathcal{L}_2(g(\boldsymbol{p}|\boldsymbol{\phi}_t)), \cdots, \mathcal{L}_m(g(\boldsymbol{p}|\boldsymbol{\phi}_t))),$$

$$\text{s.t. } \mathcal{G}_j(g(\boldsymbol{p}|\boldsymbol{\phi}_t)|\boldsymbol{p}, \boldsymbol{U}) = (\boldsymbol{u}^{(j)} - \boldsymbol{p})^T\mathcal{L}(g(\boldsymbol{p}|\boldsymbol{\phi}_t)) \leq 0, \forall j = 1, ..., K. \quad (9)$$

Since the parameter $\theta_{\boldsymbol{p}}$ is generated by the Pareto solution generator $g(\boldsymbol{p}|\phi_t)$, the generator's parameter $\phi_t$ is the only trainable parameters to be optimized. What we need is to find a valid gradient direction $\boldsymbol{d}_t$ to reduce all the losses $\mathcal{L}_i(g(\boldsymbol{p}|\phi_t))$ and activated constraints $\mathcal{G}_j(g(\boldsymbol{p}|\phi_t)|\boldsymbol{p}, \boldsymbol{U})$ with respect to $\phi_t$.

We follow the methods proposed in (Fliege & Svaiter, 2000; Gebken et al., 2017), and calculate a valid descent direction $\boldsymbol{d}_t$ by solving the optimization problem:

$$
\begin{aligned}
(\boldsymbol{d}_t, \alpha_t) = &\arg \min_{\boldsymbol{d} \in R^n, \alpha \in R} \alpha + \frac{1}{2}||\boldsymbol{d}||^2 \\
s.t. \quad &\nabla_{\phi_t} \mathcal{L}_i(g(\boldsymbol{p}|\phi_t))^T \boldsymbol{d} \leq \alpha, i = 1, ..., m \\
&\nabla_{\phi_t} \mathcal{G}_j(g(\boldsymbol{p}|\phi_t))^T \boldsymbol{d} \leq \alpha, j \in I(\phi_t).
\end{aligned}
\tag{10}
$$

where $\boldsymbol{d}_t$ is the obtained gradient direction, $\alpha$ is an auxiliary parameter for optimization, and $I(\theta) = \{j \in I | \mathcal{G}_j(g(\boldsymbol{p}|\phi_t)) \geq 0\}$ is the index set of all activated constraints. By solving the above problem, the obtained direction $\boldsymbol{d}_t$ and parameter $\alpha_t$ will satisfy the following lemma (Gebken et al., 2017):

**Lemma 1:** Let $(\boldsymbol{d}_t, \alpha_t)$ be the solution of problem (10), we either have:

1. A non-zero $\boldsymbol{d}_t$ and $\alpha_t$ with

$$
\begin{aligned}
&\alpha_t \leq -(1/2)||d_t||^2 < 0, \\
&\nabla_{\phi_t} \mathcal{L}_i(g(\boldsymbol{p}|\phi_t))^T \boldsymbol{d}_t \leq \alpha_t, i = 1, ..., m \\
&\nabla_{\phi_t} \mathcal{G}_j(g(\boldsymbol{p}|\phi_t))^T \boldsymbol{d}_t \leq \alpha_t, j \in I(\phi_t).
\end{aligned}
\tag{11}
$$

2. or $\boldsymbol{d}_t = \boldsymbol{0} \in \mathbb{R}^n$, $\alpha_t = 0$, and $\theta_{\boldsymbol{p}} = g(\boldsymbol{p}|\phi_t)$ is local Pareto critical restricted on $\Omega(\boldsymbol{p}, \boldsymbol{U})$.

In case 1, we obtain a valid descent direction $\boldsymbol{d}_t \neq \boldsymbol{0}$ which has negative inner products with all $\nabla_\phi \mathcal{L}_i(g(\boldsymbol{p}|\phi_t))$ and $\nabla_\phi \mathcal{G}_j(g(\boldsymbol{p}|\phi_t))$ for $j \in I(\phi_t)$. With a suitable step size $\eta$, we can update $\phi_{t+1} = \phi_t + \eta \boldsymbol{d}_t$ to reduce all losses and activated constraints. In case 2, we cannot find any nonzero valid descent direction, and obtain $\boldsymbol{d}_t = \boldsymbol{0}$. In other words, there is no valid descent direction to simultaneously reduce all losses and activated constraints. Improving the performance for one task would deteriorate the other(s) or violate some constraints. Therefore, the current solution $\theta_{\boldsymbol{p}} = g(\boldsymbol{p}|\phi_t)$ is a local restricted Pareto optimal solution on $\Omega(\boldsymbol{p}, \boldsymbol{U})$.

## B.2 ADAPTIVE LINEAR SCALARIZATION

As mentioned in the main paper, we can rewrite the gradient direction $\boldsymbol{d}_t$ as a dynamic linear combination with the gradient of all losses $\nabla_\phi \mathcal{L}_i(g(\boldsymbol{p}|\phi_t))$. Similar to the approach in (Fliege & Svaiter, 2000; Lin et al., 2019), we reformulate the problem (10) in its dual form:

$$
\begin{aligned}
\max_{\lambda_i, \beta_j} \quad &-\frac{1}{2}||\sum_{i=1}^m \lambda_i \nabla_{\phi_t} \mathcal{L}_i(g(\boldsymbol{p}|\phi_t)) + \sum_{j \in I(\phi_t)} \beta_i \nabla_{\phi_t} \mathcal{G}_j(g(\boldsymbol{p}|\phi_t))||^2 \\
s.t. \quad &\sum_{i=1}^m \lambda_i + \sum_{j \in I(\phi_t)} \beta_j = 1, \quad \lambda_i \geq 0, \beta_j \geq 0, \forall i = 1, ..., m, \forall j \in I(\phi_t),
\end{aligned}
\tag{12}
$$

where $\boldsymbol{d}_t = \sum_{i=1}^m \lambda_i \nabla_{\phi_t} \mathcal{L}_i(g(\boldsymbol{p}|\phi_t)) + \sum_{j \in I(\phi_t)} \beta_i \nabla_{\phi_t} \mathcal{G}_j(g(\boldsymbol{p}|\phi_t))$ is the obtained valid gradient direction, $\lambda_i$ and $\beta_i$ are the Lagrange multipliers for the linear inequality constraints in problem (10).

Based on the definition in problem (9), a constraint $\mathcal{G}_j(g(\boldsymbol{p}|\phi_t))$ can be rewritten as a linear combination of all losses:

$$
\mathcal{G}_j(g(\boldsymbol{p}|\phi_t)) = (\boldsymbol{u}^{(j)} - \boldsymbol{p})^T \mathcal{L}(g(\boldsymbol{p}|\phi_t)) = \sum_{i=1}^m (\boldsymbol{u}_i^{(j)} - \boldsymbol{p}_i) \mathcal{L}_i(g(\boldsymbol{p}|\phi_t)).
\tag{13}
$$

The gradient of the constraint is also a linear combination of those for the losses:

$$
\nabla_{\phi_t} \mathcal{G}_j(g(\boldsymbol{p}|\phi_t)) = (\boldsymbol{u}^{(j)} - \boldsymbol{p})^T \nabla_{\phi_t} \mathcal{L}(g(\boldsymbol{p}|\phi_t)) = \sum_{i=1}^m (\boldsymbol{u}_i^{(j)} - \boldsymbol{p}_i) \nabla_{\phi_t} \mathcal{L}_i(g(\boldsymbol{p}|\phi_t)).
\tag{14}
$$

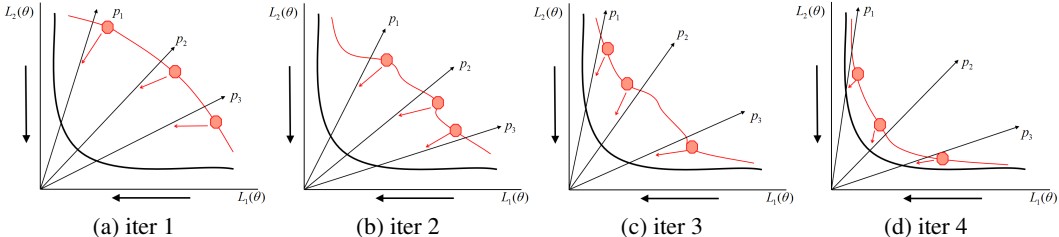

|  |  |  |  |
|---|---|---|---|
| (a) iter 1 | (b) iter 2 | (c) iter 3 | (d) iter 4 |

Figure 14: **The training process of the Pareto solution generator with multiple preferences.** At each iteration, the proposed algorithm randomly samples multiple preference vectors, and calculates a valid gradient direction which can improve the performance for all subproblems.

Therefore, we can rewrite the valid descent direction as a linear combination of the gradients for all tasks with dynamic weight $\boldsymbol{\alpha}_i(t)$:

$$\boldsymbol{d}_t = \sum_{i=1}^{m} \boldsymbol{\alpha}_i(t)\nabla_{\boldsymbol{\phi}_t}\mathcal{L}_i(g(\boldsymbol{p}|\boldsymbol{\phi}_t)), \quad \boldsymbol{\alpha}_i(t) = \lambda_i + \sum_{j\in I_{\epsilon(\theta)}} \beta_j(\boldsymbol{u}_i^{(j)} - \boldsymbol{p}_i), \tag{15}$$

where $\lambda_i$ and $\beta_j$ are obtained by solving the dual problem (12).

It should be noticed that the primal problem (10) directly optimizes the gradient direction $\boldsymbol{d}_t$, which could be in millions of dimensions for training a deep neural network. In contrast, the dimension of dual problem (12) is the number of all tasks and activated constraints (up to a few dozens), which is much smaller than the primal problem. Therefore, in practice, we obtain the valid gradient direction $\boldsymbol{d}_t$ by solving the dual problem (12) instead of the primal problem (10).

We use the Frank-Wolfe algorithm (Jaggi, 2013) to solve the problem (12) as in the previous work (Sener & Koltun, 2018; Lin et al., 2019). We use simple uniform distribution to sample both the unit preference vector $\boldsymbol{p}$ and unit reference vectors $\boldsymbol{U}$ in this paper. The number of preference vector is 1 in the previous discussion, and the number of reference vectors is a hyperparameter, which we set it as 3 in this paper. In the next subsection, we will discuss how to use more than one preference vector at each iteration to update the Pareto solution generator.

### B.3 BATCHED PREFERENCES UPDATE

To obtain an optimal Pareto generator $g(\boldsymbol{p}|\boldsymbol{\phi}^*)$, we need to find:

$$\boldsymbol{\phi}^* = \mathrm{argmin}_{\boldsymbol{\phi}}\mathbb{E}_{\boldsymbol{p}\sim P_{\boldsymbol{p}}}\mathcal{L}(g(\boldsymbol{p}|\boldsymbol{\phi})). \tag{16}$$

In the main paper, we sample one preference vector at each iteration to calculate a valid direction $\boldsymbol{d}_t$ to update the Pareto generator. A simple and straightforward extension is to sample multiple preference vectors to update the Pareto solution generator, as shown in Fig. 14.

At each iteration, we simultaneously sample and optimize multiple subproblems with respect to the Pareto generator:

$$\min_{\boldsymbol{\phi}}\{\mathcal{L}(g(\boldsymbol{p}_1|\boldsymbol{\phi})), \mathcal{L}(g(\boldsymbol{p}_2|\boldsymbol{\phi})), \cdots, \mathcal{L}(g(\boldsymbol{p}_K|\boldsymbol{\phi}))\}, \quad \boldsymbol{p}_1, \boldsymbol{p}_2, \cdots, \boldsymbol{p}_K \sim P_{\boldsymbol{p}} \tag{17}$$

where $\boldsymbol{p}_1, \boldsymbol{p}_2, \cdots, \boldsymbol{p}_K$ are $K$ randomly sampled preference vectors and each $\mathcal{L}(g(\boldsymbol{p}_k|\boldsymbol{\phi}))$ is a multi-objective optimization problem. Therefore, we now have a hierarchical multiobjective optimization problem. If we do not have a specific preference among the sampled preference vectors, the above problem can be expanded as a problem with $Km$ objectives:

$$\min_{\boldsymbol{\phi}}(\mathcal{L}_1(g(\boldsymbol{p}_1|\boldsymbol{\phi})), \cdots, \mathcal{L}_m(g(\boldsymbol{p}_1|\boldsymbol{\phi})), \cdots, \mathcal{L}_1(g(\boldsymbol{p}_K|\boldsymbol{\phi})), \cdots, \mathcal{L}_m(g(\boldsymbol{p}_K|\boldsymbol{\phi}))). \tag{18}$$

For the linear scalarization case, the calculation of valid gradient direction at each iteration $t$ is straightforward:

$$\boldsymbol{d}_t = \sum_{k=1}^{K} \nabla_{\boldsymbol{\phi}_t}\mathcal{L}(g(\boldsymbol{p}_k|\boldsymbol{\phi}_t)) = \sum_{k=1}^{K}\sum_{i=1}^{m} \boldsymbol{p}_i\nabla_{\boldsymbol{\phi}_t}\mathcal{L}_i(g(\boldsymbol{p}_j|\boldsymbol{\phi}_t)), \quad \boldsymbol{p}_1, \boldsymbol{p}_2, \cdots, \boldsymbol{p}_K \sim P_{\boldsymbol{p}}, \tag{19}$$

where we assume all sampled subproblems are equally important.

It is more interesting to deal with the preference-conditioned multiobjective optimization problem. For each preference vector $\boldsymbol{p}_k$, suppose we use the rest preference vectors as its reference vectors, the obtained preference-conditioned multiobjective optimization problem would be:

$$\min_{\boldsymbol{\phi}_t}(\mathcal{L}_1(g(\boldsymbol{p}_1|\boldsymbol{\phi}_t)), \cdots, \mathcal{L}_m(g(\boldsymbol{p}_1|\boldsymbol{\phi}_t)), \cdots, \mathcal{L}_1(g(\boldsymbol{p}_K|\boldsymbol{\phi}_t)), \cdots, \mathcal{L}_m(g(\boldsymbol{p}_K|\boldsymbol{\phi}_t)))$$

$$\text{s.t. } \mathcal{G}_{1j}(g(\boldsymbol{p}_1|\boldsymbol{\phi}_t)) = (\boldsymbol{p}_j - \boldsymbol{p}_1)^T \mathcal{L}(g(\boldsymbol{p}_1|\boldsymbol{\phi}_t)) \le 0, \forall j \in \{1, ..., K\}\backslash\{1\},$$

$$\mathcal{G}_{2j}(g(\boldsymbol{p}_2|\boldsymbol{\phi}_t)) = (\boldsymbol{p}_j - \boldsymbol{p}_2)^T \mathcal{L}(g(\boldsymbol{p}_2|\boldsymbol{\phi}_t)) \le 0, \forall j \in \{1, ..., K\}\backslash\{2\}, \tag{20}$$

$$\cdots$$

$$\mathcal{G}_{Kj}(g(\boldsymbol{p}_K|\boldsymbol{\phi}_t)) = (\boldsymbol{p}_j - \boldsymbol{p}_K)^T \mathcal{L}(g(\boldsymbol{p}_K|\boldsymbol{\phi}_t)) \le 0, \forall j \in \{1, ..., K\}\backslash\{K\}.$$

There are $(K - 1)$ constraints for each preference vector, hence total $K(K - 1)$ constraints for the preference-conditioned multiobjective problem, although some of them could be inactivated. Similar to the single point case, we can also calculate the valid gradient direction in the form of adaptive linear combination as:

$$\boldsymbol{d}_t = \sum_{k=1}^{K}\sum_{i=1}^{m} \boldsymbol{\beta}_i(t)\nabla_{\boldsymbol{\phi}_t}\mathcal{L}_i(g(\boldsymbol{p}_k|\boldsymbol{\phi}_t)), \tag{21}$$

where the adaptive weight $\boldsymbol{\beta}_i(t)$ depends on all loss functions $\mathcal{L}_i(g(\boldsymbol{p}_k|\boldsymbol{\phi}_t))$ and activated constraints $\mathcal{G}_{kj}(g(\boldsymbol{p}_k|\boldsymbol{\phi}))$.

## C    EXPERIMENTAL SETTING

**Synthetic Example:**    The synthetic example we use in Section 6 is defined as:

$$
\begin{aligned}
\min_{\boldsymbol{\theta}} f_1(\boldsymbol{\theta}) &= 1 - \exp\left(-(\boldsymbol{\theta}_1 - 1)^2 - \frac{1}{n-1}\sum_{i=2}^{n}[\boldsymbol{\theta}_i - \sin(5\boldsymbol{\theta}_1)]^2\right), \\
\min_{\boldsymbol{\theta}} f_2(\boldsymbol{\theta}) &= 1 - \exp\left(-(\boldsymbol{\theta}_1 + 1)^2\right).
\end{aligned}
\tag{22}
$$

We set $n = 10$ and use a simple two-layer MLP network with 50 hidden units on each layer to generate the Pareto solutions based on the preference vectors.

**MultiMNIST (Sabour et al., 2017):**    In this problem, the goal is to classify two overlapped digits in an image at the same time. The size of the original MNIST image is $28 \times 28$. We randomly choose two digits from the MNIST dataset, and move one digit to the upper-left and the other one to the bottom right with up to $4$ pixels. Therefore, the input image is in size $36 \times 36$.

Similar to the previous work (Sener & Koltun, 2018; Lin et al., 2019), we use a LeNet-based neural network with task-specific fully connected layers as the main MTL model, and use a simple MLP as the hypernetwork. Since the LeNet model is small, we do not use any preference/chunk embedding. We let the hypernetwork-based model has a similar number of parameters with a single MTL model. For all methods, the optimizer is Adam with learning rate lr $= 3e^{-4}$, the batch size is 256, and the number of epochs is 200.

**CityScapes (Cordts et al., 2016):**    This dataset has street-view RGB images, and involves two tasks to be solved, which are pixel-wise semantic segmentation and depth estimation. We follow the setting used in (Liu et al., 2019), and resize all images into $128 \times 256$. For the semantic segmentation, the model predicts the 7 coarser labels for each pixel. We use the L1 loss for the depth estimation. We report the experimental results on the Cityscapes validation set.

We use the MTL network proposed in (Liu et al., 2019), which has SegNet (Badrinarayanan et al., 2017) as the shared representation encoder, and two task-specific lightweight convolution layers. In our hypernetwork-based model, the hypernetwork contains three 2-layer MLPs with 100 hidden units on each layer, and most parameters are stored in the parameter tensors for linear projection. The preference embedding and chunk embedding are all 64-dimensional vectors. We also let the hypernetwork-based model has a similar number of parameters with the single MTL model. For all experiments, we use Adam with learning rate lr $= 3e^{-4}$ as the optimizer, and the batch size is 12. We train the model from scratch with 200 epochs.

**NYUv2 (Silberman et al., 2012):**    This dataset is for indoor scene understanding with two tasks: a 13-class semantic segmentation and indoor depth estimation. Similar to Liu et al. (2019), we resize all images into $288 \times 384$. For the training from scratch experiment, we use a similar MTL network and hyperparameter setting as for the CityScapes problem, except the batch size is 8 in this problem.

We also test the performance on models with pretrained encoder on the NYUv2 Dataset. We follow the setting in the recent MTL survey paper (Vandenhende et al., 2020), all models have a ResNet-50 backbone (He et al., 2016) pretrained on ImageNet, and two-specific heads with ASPP module (Chen et al., 2018a). In our hypernetwork-based model, the hypernetwork has three different 2-layer MLPs with 200 hidden unit on each layer. One MLP is for generating shared convolution layers on top of the ResNet backbone, and the other two are for each task-specific head. The shared parameters for backbone are unfrozen, and will be adapted during training. We use 100-dimensional vectors as the preference and chunking embedding. We use Adam with learning rate lr $= 1e^{-4}$ as the optimizer, the batch size is 8, and the total epoch is 200.

**CIFAR100 with 20 Tasks:**    To validate the algorithm performance on MTL problem with many tasks, we split the CIFAR-100 dataset (Krizhevsky & Hinton, 2009) into 20 tasks, where each task is a 5-class classification problem. Similar setting has been used in MTL learning (Rosenbaum et al., 2018) and continual learning (von Oswald et al., 2020).

The MTL neural network has four convolution layers as the shared architecture, and 20 task-specific FC layers. In our proposed model, we have 5 MLPs and the preference and chunking embedding are both 32-dimensional vectors. The optimizer is Adam with learning rate lr $= 3e^{-4}$, the batch size is 128, and the number of epochs is 200. We report the test accuracy for all 20 tasks.

