# OpenReview forum: "Controllable Pareto Multi-Task Learning"
_ICLR.cc/2021/Conference — Reject_

### Official Review · AnonReviewer1 · 2020-10-29
**The proposed technique does not match with the announced contribution.**

**Rating:** 4
**Confidence:** 5

**Review:**

This paper proposes a novel controllable Pareto multi-task learning framework, which aims to learn the whole Pareto optimal front for all tasks with a single model. The motivation is straight forward and the proposed method is inspiring. However, the proposed technique does not match with the announced contribution.
1. This paper announces that it proposes a novel Pareto solution generator that can learn the whole Pareto front for MTL. However, actually, this paper adopts fixed shared parameters (pretrained feature extractor) and only optimizes a part of parameters of a MTL, which degrade the solution space and the solutions may not be the Pareto solutions. The Pareto front generated by such method may not be the real Pareto front for MTL.
2. Using fixed shared parameters conflicts with the essence Pareto MTL. In MTL, the tasks mutually regularized by the feature extractor, which improve the generalization ability of each task.

This paper is globally well organized and clearly written. However, some important details are missing.
1. The details about the hypernetwork are unclear.
2. The paper lacks of analysis on the experimental result.
3. Some notations are not clear, e.g., does the loss used in the paper denotes empirical loss?

---

> ### Author Response · Authors · 2020-11-12
> **Response to R1 Part1**
>
> Thank you for your efforts in reviewing our paper. We provide our responses to your concerns point by point as follow:
>
> > Fixed Shared Parameters
>
> We believe there are some misunderstandings on the model we proposed in this paper. We want to clarify them here and will revise our paper to make these points clear:
>
> 1. For all experiments in the current paper, we **do not use any pretrained feature extractor**, and all multi-task deep neural network (including our proposed hypernetwork-based model) are **trained from scratch**. In our proposed method, we use the hypernetwork to **generate all parameters** for the main MTL network. With the model compression ability powered by the hypernetwork (like chunking [1,2]), our hypernetwork-based model has a similar number of trainable parameters with the traditional MTL network (e.g., both have 26 million parameters for the SegNet-based MTL model in the CityScape and NYUv2 problems, with the same structure as in [3]). ~~We will make this point clear, add details of our hypernetwork-based model in the revised manuscript.~~ We have added the details of our hypernetwork-based model in Section 5 to make this point clear. We will also open-source the code for all experiments, to let other researchers easily reproduce our results by training the models from scratch (see code examples in supp.zip).
>
> 2. The misunderstanding might also come from the parameter sharing method (e.g., shared frozen or unfrozen pretrained feature extractor) we discussed in Appendix C (Fig. 16(b) in the original manuscript). This is the technique which can further improve the scalability of our proposed method, but it is **not used** in ~~this paper~~ the original manuscript. The use of pretrained feature extractor (on a large scale dataset) can improve the performance of an MTL model (see a recent survey[4]). In our proposed method, such a shared feature extractor (frozen or unfrozen) can be treated as a structure constraint across all generated solutions. As correctly pointed out by the reviewer, with the shared feature extractor ~~(which is not used yet in the paper)~~, our method will generate a constrained Pareto front but not the real (unconstrained) Pareto front. We ~~will add~~ have added discussions and experiments in the revised manuscript.
>
> 3. Since the objective functions for a multi-task neural network could be highly non-convex, the best solutions we can find is the Pareto stationary solutions rather than the global Pareto optimal solutions (similar to the local/stationary optimal solution for single-objective a deep neural network). It is also the case for all current work. More details can be found in response to AnonReviewer3 above. We ~~will carefully revise~~ have carefully revised our announced contribution accordingly in the manuscript.
>
> > MTL as Multi-Objective Optimization
>
> 1. The idea of reformulating the MTL problem as multiobjective optimization is a timely research direction, and have drawn attention from researchers in the MTL community[5-8]. The current works directly use existing methods, and can only find a single [5] or a finite set of Pareto stationary solutions for an MTL problem [6-8]. In addition, these methods need to train and store a large number of deep neural networks with different preferences, which is expensive and prohibited in many real-world applications.
>
> 2. In contrast, we propose a novel method to use a single model to learn the whole trade-off curve (which might contain infinite solutions with different preferences). Thanks to our proposed end-to-end training procedure, the hypernetwork-based model can simultaneously learn to generate the corresponding Pareto stationary solutions for different preferences. In other words, our proposed method provides an efficient way to solve MTL problems with different trade-off preferences by a single model. There is a concurrent ICLR submission shares similar idea with our work(https://openreview.net/forum?id=NjF772F4ZZR). They also demonstrate the benefit of reformulating MTL as solving multiobjective optimization problem, and propose to use hypernetwork to learn the entire trade-off curve. We ~~will add~~ have added discussion with this work in our revised manuscript.

---

> > ### Author Response · Authors · 2020-11-12
> > **Response to R1 Part2**
> >
> > > Regularization
> >
> > 1. We agree with you that regularization is important to improve the generalization ability. Unlike the traditional MTL model with a fixed shared feature extractor, our proposed method puts the regularization on the hypernetwork. Conceptually, there are two steps to obtain an output from our hypernetwork-based model. In step 1, with a specific preference, the hypernetwork will generate all parameters for the main MTL net, which has the same structure (e.g., shared bottom + task-specific heads) as a traditional MTL net. In step 2, an input $x$ to the main MTL net will first go through the shared bottom (feature extractor with parameters generated by the hypernetwork) and then all task-specific heads (also with parameters generated by the hypernetwork) to obtain the output vector $\\{y_1,y_2,...,y_m\\}$, which is similar to the traditional MTL model. In this way, all tasks are still regularized by the shared feature extractor, of which all parameters are now generated by the hypernetwork. We ~~will revise~~ have revised the Fig.4 to make this point clear and ~~add~~ have added discussion in the revised manuscript.
> >
> > 2. There is an extra implicit regularization effect across different trade-off preferences. Our goal is to train a single hypernetwork-based model that performs well for all preferences. In other words, the generated main MTL network with a specific preference is regularized by the other networks with different preferences through the hypernetwork. This regularization effect might be best explained by the batched preferences update we discussed in Appendix ~~A.4~~ B.3. In this respect, it is similar to [7] uses hypernetwork with explicit regularization on separate tasks in the continual learning setting. It is good for knowledge transfer among the preference-conditioned networks.
> >
> > 3. With this regularization and knowledge transfer among different preferences, our proposed method can learn the trade-off curves pretty well by a single model on various MTL problem as shown in ~~Fig.7 and 8.~~ Fig. 8-10.
> >
> > > Missing Details
> >
> > 1. We ~~will add~~ have added the details of our hypernetwork-based model with the discussions above to the revised manuscript.
> >
> > 2. More experimental results, along with analysis ~~will be added~~ have been added in the paper.
> >
> > 3. Yes, we use empirical loss to train our proposed hypernetwork-based model. We ~~will carefully revise~~ have revised the paper to make this point clear.
> >
> > We hope these responses can address your concerns. Please let us know if you have any questions and further concerns. We look forward to additional discussion.
> >
> > [1] Ha, et al. Hypernetworks. ICLR 2017.
> >
> > [2] von Oswald, et al. Continual learning with hypernetworks. ICLR 2020.
> >
> > [3] Liu, et al. End-to-end multi-task learning with attention. CVPR 2019.
> >
> > [4] Vandenhende, et al. Multi-Task Learning for Dense Prediction Tasks: A Survey. arxiv: 2004.13379.
> >
> > [5] Sener \& Koltun. Multi-task learning as multi-objective optimization. NeurIPS 2018.
> >
> > [6] Lin et al. Pareto multi-task learning. NeurIPS 2019.
> >
> > [7] Ma et al. Efficient continuous Pareto exploration in multi-task learning. ICML 2020.
> >
> > [8] Mahapatra \& Rajan. Multi-task learning with user preferences: Gradient descent with controlled ascent in pareto optimization. ICML 2020.

---

> ### Author Response · Authors · 2020-11-23
> **Updated Response to R1**
>
> We have updated a revised version of our manuscript where the summary can be found in the above post. Here we want to provide you a point-to-point summary related to your concerns:
>
> Major Concerns
>
> 1. **Model Structure**: We want to emphasize that our hypernetwork-based model generates **all parameters for the MTL model** but not just a part of it, which are the cases for all experiments in the original paper. More details of the proposed model have now been added in Section 5 to make this point clear. Our proposed model can optionally incorporate pretrained feature extractors into our model, which makes it a constrained multiobjective optimization problem. We have added a new experiment on model with ResNet-50 pretrained backbone, and provided more discussion in Appendix A. We will also open source our code to let other researchers easily reproduce our results (see supp.zip for code examples).
>
>
> 2. **Contribution**: Using multi-objective optimization methods to solve MTL is a timely research direction, and a few works have been recently proposed to find a single or a set of finite Pareto stationary solutions for an MTL problem[5-8]. Along this research line, we propose to learn the whole front of Pareto stationary solutions with a single model, which could be a novel contribution to the MTL community. A concurrent ICLR submission also proposes a similar idea with us (https://openreview.net/forum?id=NjF772F4ZZR).
>
>
> 3. **More on Contribution**: Even though our proposed model can learn the trade-off curve, it only has a comparable size with a single MTL model. The proposed model can **scale well for large scale problems and supports real-time trade-off preference adjustment for inference**. It provides a novel and efficient way to solve MTL problems, and could be a valuable contribution to the MTL community.
>
>
> 4. **Regularization**: In our proposed method, the main MTL model always has the same (shared feature extractor --> task-specific heads) structure, and only the parameters are generated by the hypernetwork. The data input to the generated MTL model will first go through the shared feature extractor, and then task-specific heads to obtain different output ys. In other words, different tasks are still regularized by each other via the feature extractor. Our model put the regularization effect on the hypernetwork, which should generate parameters that work well for all tasks, given different preferences. We have added a discussion on the regularization issue in Section 5.
>
> Important Details:
>
> 1. **Hypernetwork**: We have added more details about the hypernetwork in Section 5. The details (e.g., hyperparameters) on the specific networks we used for each problem are provided in Appendix C.
>
>
> 2. **Experiments**: We have added more analysis for the experimental results. In addition, we also added a new experiment with pretrained ResNet-50 backbones. Our model can scale well for a large model size from 23M (train from scratch, generate all parameters) to more than 100M (with pretrained backbone), and has a relatively small latency overhead to support real-time trade-off adjustment.
>
>
> 3. **Notations**: We have carefully revised the manuscript to make the notations clear, thank you for pointing this out.
>
> We hope this revision and responses can address all your concerns. A detailed initial response can be found below. Please let us know if you have further concerns and questions on our work. Thank you again for reviewing our paper.

---

### Official Review · AnonReviewer4 · 2020-10-29
**Multi-Objective Optimization for Multi-Task Learning**

**Rating:** 7
**Confidence:** 4

**Review:**

The paper proposes a method to controllably generate models on the trade-off front of multi-task learning problems. The key idea is to use a hypernetwork that will generate the MTL parameters on demand conditioned on the desired trade-off. The hypernetwork can be trained along with the MTL in an end-to-end manner.

Strengths:
+ Paper addresses an important practical problem. Instead of training many models independently to span the trade-off front, it can generate an MTL model on demand for any desired trade-off point.
+ The idea of using a hypernetwork in the context of MTL is new.

Weaknesses:
- Conceptually and technically the improvements over [1] are minimal. The main contribution of the paper is to use a hypernetwork that will generate the weights. Although this is new in the context of MTL, it is in fact a common idea in the context of Neural Architecture Search. In NAS we have a supernet which is equivalent to the hypernet and then perform reference based multi-objective optimization over it. See [2] for an example.
- Method does not seem to be very scalable in terms of MTL model size. The experimental evaluation is on a small scale. For larger models the hypernet needs to generate a large output for the desired trade-off. In that case training the hypernet gets more challenging.
- In terms of the exposition, firstly the main paper has very few details apart from the high level idea, there is a lot of repetition of the same points. Secondly, a lot of the actual paper and the appendix itself is background material that is already well known in the multi-objective optimization literature.
- The experiments are a bit disappointing. The advantage of the proposed method in comparison to linear scalarization would be on concave pareto fronts as shown in Fig. 6. Unless I am missing something, looks like in none of the actual experiments is the trade-off a concave curve.

Overall, in the current form the paper looks more like a proof of concept. I would encourage the authors to demonstrate or discuss the scalability of the solution.

Clarifications:
1) I guess one of the benefits of this approach over training multiple models is in terms of the total number of parameters and computational complexity. But there is no discussion of these aspects. How does the HyperNet compare in size wrt to the MTL model part? I would imagine the HyperNet is much bigger since it has to learn to predict the MTL parameters.
2) How about inference? How long does it take for the HyperNet to generate the parameters since you still have to solve an optimization problem?
3) The paper claims real-time but there is no discussion of this claim.
4) In Fig.7 there seem to be some solutions that are better than the trade-off front obtained by the proposed method. Any comments on what is limiting the current approach from reaching/surpassing those solutions?

[1] Pareto Multi-Task Learning
[2] Neural Architecture Transfer

---

> ### Author Response · Authors · 2020-11-12
> **Response to R4 Part1**
>
> Thank you for your efforts in reviewing our paper. We provide our responses to your concerns point by point as follow:
>
> > Improvements and Contributions
>
> 1. We totally agree with the reviewer that the use of hypernetwork to generate the weights for the main network itself is not a new idea, and our contribution is to use it to generate the whole trade-off curve for MTL.
>
> 2. We think there are some critical misunderstandings on the scalability and the inference behavior of our work. In short, our hypernetwork-based model has **a similar number of parameters with the MTL model**, and can scale well for large models. Thanks to the end-to-end training, our model **does not need extra optimization steps at the inference time**. We will address these two important concerns in detail in the following response and hope they can make our contribution much clearer. We ~~will also revise~~ have revised our manuscript carefully to avoid misunderstanding.
>
> 3. The idea of using multiobjective optimization methods to solve MTL is a timely research direction, and has drawn attention from researchers in the MTL community [1,3-5]. Learning the entire trade-off curve, instead of a finite set of separate solutions, would be a novel contribution to multiobjective optimization, and also an important improvement over the current related works that use multiobjective optimization to solve MTL problem. We find there is a concurrent ICLR submissions that shares similar idea with our work (https://openreview.net/forum?id=NjF772F4ZZR) to learn the entire Pareto front by hypernetwork. We ~~will add~~ have added discussions with this work in our manuscript. See above response to AnonReviewer3 for more detailed discussions on the contribution.
>
> 4. Thank you for pointing out the closely related work (NAT) [2], which is a novel and efficient MO-NAS method. We have cited and discussed this work in the revised paper. NAT simultaneously optimizes a set of neural networks, which are sampled from a large supernet, to approximate the optimal trade-off curve among different objectives via a single run. The obtained supernet also supports fast adaption to new tasks and domains.  Our proposed method is different from NAT, and it learns a direct mapping from a specific preference to the corresponding Pareto stationary solution without any search/selection/optimization, which is good for real-time trade-off adjustment.  Our proposed method might further improve the preference-based fast adaption ability for the current NAS methods, and searching different architectures for different preferences is also an important improvement for MTL.
>
> >  Scalability and Experiments
>
> 1. Scalability: With the model compression ability powered by the hypernetwork (e.g., chunking[7,8]), our proposed hypernetwork-based model can scale well to large scale problems. In this paper, our proposed model has a comparable number of parameters with the corresponding single MTL model in all experiments. For example, for the CityScape and NYUv2 experiments, we use the SegNet-based MTL model proposed in [6], and both hypernetwork-based model and a single MTL model have around 26 million parameters. Given the current works [1,4,5] need to train and store multiple MTL models (say, 10 models, total 260M parameters), our proposed model is much more parameter-efficient while learning the whole trade-off curve. In this sense, it is much more scalable than the current methods which need to train multiple MTL models.
>
> 2. End-to-End Traning: Thanks to the proposed end-to-end gradient-based optimization method, training our model only requires the same number of optimization iterations (epochs) to train a single traditional MTL network, while our model can learn the entire trade-off curve. In other words, our proposed method provides an efficient way to solve MTL problems with different trade-off preferences by a single model.
> The properties of multiobjective optimization, such as the trade-off curve is a low dimensional manifold, and similar preference vectors have similar hypernetwork outputs, make it possible to learn the curve to cover different (infinite) trade-off preferences with a single model. Simultaneously training the hypernetwork-based model with different preferences is also good for cross-preference regularization. See the below responses to AnonReviewer1 on model structure and regularization.
>
> 3. Experiments: We follow similar experimental settings (datasets, problems and models) with the most related current works[1,3-6]. The model size is up to 26 million parameters, and the number of tasks is up to 20. We ~~will add~~ have added more experimental results with analysis to demonstrate the scalability of our proposed method (e.g., model with pretrained feature extractor, up to 100M parameters) in the revised manuscript. We will also open-source the code for all experiments to let other researchers easily reproduce our results (see code examples in supp.zip).

---

> > ### Author Response · Authors · 2020-11-12
> > **Response to R4 Part2**
> >
> > > Inference and Real-Time
> >
> > 1. Inference: Similar to other hypernetwork-based models [7,8], we **do not need to solve any optimization problem at the inference time**. The hypernetwork takes the preference vector as input, and directly generate the parameters for the main MTL network. We notice that some hypernet(supernet)-based methods for NAS have a bilevel optimization process, where extra optimization steps are needed to obtain the suitable parameters (inner optimization)[9,10]. This is not the case for our hypernetwork-based model, where it can directly generate the preference-conditioned parameters for inference.
> >
> > 2. Real-Time: Conceptually, there are two steps to use the hypernetwork-based model at the inference time: 1) the hypernet takes the preference as input to generate the parameter for the main MTL network, and then 2) the main MTL network takes the data input and predict the outputs. Since the hypernetwork-based model has a similar number of parameters with the MTL model, it will not have a much larger inference time. That is why we claim our proposed model can conduct preference adjustment in real-time (without extra optimization steps). In our experiments, we do not observe a significant computational overhead to conduct inference with different preference vectors. We have added a discussion on scalability and fast inference in Section 5. We have also reported and analyzed the model size and inference latency for all models in the experiment section.
> >
> >
> > > Exposition
> >
> > 1. Thank you for pointing this out. We ~~will carefully revise~~ have carefully revised the manuscript to make it concise and to the point. More details and discussion on the hypernetwork-based model (e.g., the model structure, scalability, and inference) ~~will be~~ are now provided in the main paper.
> >
> > 2. We agree with the reviewer that some material in this paper is well-known in the multiobjective optimization community. ~~However, we also find it valuable to put them in the context of MTL to make the paper clear and self-contained, especially for readers with an MTL background but unfamiliar with multi-objective optimization. We will first move some material on multiobjective optimization from main paper to the appendix, to make room for details and more discussions on the proposed hypernetwork-based model and experimental analysis. Any suggestion is welcome for better organizing the paper. ~~ We have carefully revised the paper to make the discussion on multiobjective optimization concise and to the point. More details and discussions on the proposed model, as well as experimental analysis, have been added in the main paper. We have also reorganized the appendix, to let it only keep the most relevant discussion.
> >
> > 3. Many components of the optimization method we use are well-known in the multiobjective optimization literature to find a single Pareto stationary solution. With our proposed method, they can be easily generalized to find the entire trade-off curve. We believe this is an advantage, since it has the potential to incorporate more (advanced) multiobjective optimization method to learn the trade-off curve. We hope our work can further bridge the research on multiobjective optimization and multi-task learning, along with the current related works[1,3-5].
> >
> > > Concave Front
> >
> > 1. The linear scalarization method cannot find any point on the concave part of the Pareto front (not just bad for entire concave Pareto fronts). It can find the whole Pareto front only when the entire Pareto front is convex. In Fig ~~6~~7, since the ground truth Pareto front is an entire concave curve, the linear scalarization method can only find the two endpoints.
> >
> > 2. It becomes more interesting to analyze the performance of real-world MTL problems. Since the objective functions for a deep multi-task neural network could be highly non-convex, the optimization landscape would be extremely complicated. We do not know the ground truth Pareto front for the real-world MTL problems (and whether it is convex/concave or mixed), which is similar to we do not know the single best solution for a single objective deep neural network on real-world problems, as discussed in the above response to AnonReviewer4.
> >
> > 3. However, as reported in the current related works on finding separate Pareto solutions for an MTL problem [1,3-5], as well as in our experiments, the linear scalarization method always has worse performance compared with the multiobjective optimization methods. One possible explanation could be the "always attracting by the convex part of the (local) Pareto front behavior" hinders the optimization process of the linear scalarization method.

---

> > > ### Author Response · Authors · 2020-11-12
> > > **Response to R4 Part3**
> > >
> > > > Better Performance on Some Single Solution
> > >
> > > 1. As discussed above, our hypernetwork-based model has a similar number of parameters with the corresponding MTL model, and it needs to use the single model to learn the entire Pareto curve. Although the nature of model compression and cross-preference regularization might improve the generalization performance, the limited learning capacity might sometimes make some parts of the trade-off front dominated by a single solution. ~~We will compare the performance of our proposed model with different sizes in the revised manuscript. Some preliminary results show that increasing the model size can improve our model's learning capacity and raise performance.~~ Simply increasing the model size can only slightly improve our model's performance, but will lead to a large inference latency overhead which is not good for real-time preference adjustment. Advanced task balancing methods and/or better architecture designs would be useful to further improve the performance and inference latency.
> > >
> > > 2. Even with the same model size, our generated trade-off front still provides a large part of non-dominated solutions with different trade-off preferences to the single solution counterpart. In other words, no single solution is strictly better than the trade-off front. In Fig.~~7~~8, the single-task baselines always have 2X number of parameters (two full models for two tasks), which make it non-dominated by our model.
> > >
> > > We hope these responses can address your concerns. Please let us know if you have any questions and further concerns. We look forward to additional discussion.
> > >
> > > [1] Lin et al. Pareto multi-task learning. NeurIPS 2019.
> > >
> > > [2] Lu et al. Neural Architecture Transfer. arxiv: 2005.05859
> > >
> > > [3] Sener \& Koltun. Multi-task learning as multi-objective optimization. NeurIPS 2018.
> > >
> > > [4] Ma et al. Efficient continuous Pareto exploration in multi-task learning. ICML 2020.
> > >
> > > [5] Mahapatra \& Rajan. Multi-task learning with user preferences: Gradient descent with controlled ascent in pareto optimization. ICML 2020.
> > >
> > > [6] Liu, et al. End-to-end multi-task learning with attention. CVPR 2019.
> > >
> > > [7] Ha, et al. Hypernetworks. ICLR 2017.
> > >
> > > [8] von Oswald, et al. Continual learning with hypernetworks. ICLR 2020.
> > >
> > > [9] Brock, et al. Smash: One-shot model architecture search through hypernetworks. ICLR 2018
> > >
> > > [10] MacKay, et al. Self-tuning networks: Bilevel optimization of hyperparameters using structured best-response functions. ICLR 2019.
> > >
> > > [11] Vandenhende, et al. Multi-task learning for dense prediction tasks: A survey. arxiv: 2004.13379

---

> ### Author Response · Authors · 2020-11-23
> **Updated Response to R4:**
>
> We have updated a revised version of our manuscript where the summary can be found in the above post. Here we want to provide you a point-to-point summary related to your concerns:
>
> 1. **Contribution**:  By learning the trade-off curve, our model provides a direct mapping from a specific preference to the corresponding Pareto stationary solution without any search/selection/optimization, which makes it a novel way to solve MTL as multi-objective optimization problem. In addition, our proposed hypernetwork design make it scale well for large model and support real-time trade-off adjustment, which is valuable for many real-world applications. We believe this contribution would also be useful for other multiobjective optimization applications such as MO-NAS.
>
>
> 2. **Scalability**: Our model has a comparable size with a single MTL model, so it can scale well for large model size. We have added a detailed discussion on the proposed model in Section 5 to make this point clear. Compared with training a single MTL model, our proposed end-to-end training method also only need the same number of update steps (epochs) for training.
>
>
> 3. **Exposition**: Thank you for this valuable comment. We have carefully revised the paper to make the discussion on multiobjective optimization concise and to the point. More details and discussions on the proposed model, as well as experimental analysis, have been added in the main paper. We have also reorganized the appendix, to let it only keep the most relevant discussion. We believe our paper now has a much better exposition.
>
>
> 4. **Experiments**: We emphasize that our original training from scratch experiments follow similar settings (problems/models) with the most related work on MTL[1,3-6]. The model size is up to 26M, and the number of tasks is up to 20. We add a new experimental comparison on MTL models with pretrained ResNet-50 backbone, following the setting in a recent MTL survey paper[11]. Our model in this experiment has more than 100M parameters, which demonstrate the scalability of our proposed model. In contrast, a concurrent work share the similar idea (https://openreview.net/forum?id=NjF772F4ZZR) only has models with up to 0.37M parameters.
>
>
> 5. **Model size and Inference**: We provide the model size and latency of all models we used in the experiment section, along with proper discussion. The comparable model size and reasonable latency overhead make the proposed hypernetwork scale well for large model size and support real-time trade-off adjustment for inference. We provide detailed experimental settings (e.g., hyperparameters) for all problems in the appendix.  We will also open source our code to let other researchers easily reproduce our results (see supp.zip for code examples).
>
> We hope this revision and responses can address all your concerns. A detailed initial response can be found below. Please let us know if you have further concerns and questions on our work. Thank you again for reviewing our paper.

---

### Official Review · AnonReviewer3 · 2020-10-30
**a paper with overstatements**

**Rating:** 5
**Confidence:** 4

**Review:**

This paper proposes a controllable Pareto multi-task learning model by generating the Pareto-stationary solutions.

Even though the idea to generate a Pareto-stationary solution seems interesting, the proposed method is overstated. It is well known that the MGDA can find Pareto-stationary solutions, which however are NOT Pareto-optimal solutions. The steepest gradient descent method to solve problem (10) is just the primal form of the MGDA method and hence it cannot find the Pareto-optimal solution. To the best of my knowledge, there is no method which can guarantee to find a Pareto-optimal solution for general multi-objective optimization problems. Authors claim that their method can generate Pareto-optimal solutions but actually they can generate only Pareto-stationary solutions at most. Authors confuse the Pareto-optimum with Pareto-stationary solutions and this is misleading. In this sense, the proposed method is not so appealing as Pareto-stationary solutions are easy to obtain in many methods.

---

> ### Author Response · Authors · 2020-11-12
> **Response to R3 Part1**
>
> Thank you for your efforts in reviewing our paper. We provide our responses to your concerns point by point as follow:
>
> > Pareto-stationary solution
>
> 1. As you correctly point out, the solutions our method can generate are Pareto stationary solutions rather than global Pareto optimal solutions. The Pareto stationary is a necessary condition for Pareto optimality [1,2], and more strong requirements are needed to make it a sufficient condition (e.g., all functions are convex and all $\lambda$  are strictly positive in the dual problem) [1,3]. For a general multiobjective optimization problem, no method can guarantee to find a Pareto optimal solution. We will modify our statements accordingly and carefully revise our manuscript to make this point clear.
>
> We believe our proposed method is useful for solving the MTL problem with deep neural networks:
>
> 2. This Pareto stationary v.s. Pareto optimal situation is similar to the well-known stationary/local optimal v.s. global optimal situation for training a single-objective deep neural network. The objective function of a deep neural network could be highly non-convex, and the gradient-based methods (e.g., SGD and Adam) can not guarantee to find a global optimal solution. However, these methods are widely-used to train deep neural networks, and achieve promising performance in practice.  The experimental results show that our proposed method can generate reasonable trade-off curves for different MTL problems, which could be useful for many real-world MTL applications.
>
> 3. Non-convex optimization on deep neural networks [4-6] is an important and on-going research topic. How to generalize the results from single-objective optimization to multiobjective optimization would be an interesting research direction in future work.
>
> > Contribution: "The proposed method is not so appealing as Pareto-stationary solutions are easy to obtain in many methods."
>
> 1. We think there is a misunderstanding here. Instead of obtaining a single or a set of finite Pareto stationary solutions by other multiobjective optimization methods, we propose to **learn the entire trade-off curve for a given multiobjective optimization problem (which might contain infinite Pareto stationary solutions) by a single model**. The result we provide to the user/decision-maker is the model but not a set of finite Pareto stationary solutions. By adjusting the preference vector, decision-makers can obtain their preferred Pareto stationary solution(s) in real-time, rather than rerunning the optimization algorithm if none of the finite solutions satisfy their preference. This is a novel contribution for solving multiobjective optimization problems, and could be valuable for many real-world applications. (There are a few concurrent works, see 5).
>
> 2. Under mild conditions, the Pareto set/front for a continuous multiobjective optimization problem is an (m-1)-dimensional manifold in the solution/objective space [7]. In our proposed methods, the valid set of preference vector is also an (m-1)-dimensional manifold, where m is number of tasks. Our proposed hypernetwork-based model provides a natural mapping from the preference vector to the trade-off curve and supports an efficient end-to-end gradient-based method to train the model. By going through the preference vectors, we can reconstruct the entire trade-off curve as shown in our experiments (e.g., Fig 7,8), which provides an intuitive and efficient way to analyze the trade-off among different tasks. The proposed model structure is also a novel contribution of this paper.

---

> > ### Author Response · Authors · 2020-11-12
> > **Response to R3 Part2**
> >
> > 3. The idea of using multiobjective optimization methods to solve MTL is a timely research direction, and has drawn attention from researchers in the MTL community. Some works have been reported in the recent ml conferences [8-11]. **These current works directly use existed optimization methods, and can only find a single [8] or a finite set of Pareto stationary solutions for an MTL problem [9-11].** In addition, these methods need to train and store a large number of separate deep neural networks (each model might have millions or even billions of parameters) for different preferences, which is expensive and prohibited in many real-world applications. In contrast, we propose a novel method to use a single model to learn the whole trade-off curve (which might contain infinite solutions), which allows the users to obtain solutions with their preferred trade-off in real time. This real-time adjustment ability is valuable to many real-world MTL applications.
> >
> > 4. Thanks to the model compression ability powered by hypernetwork, **our hypernetwork-based model only has a comparable number of parameters with a single traditional MTL network.** In addition, training our model only requires the same number of optimization iterations (epochs) as to train a single traditional MTL network, while our model can learn the entire trade-off curve. In other words, our proposed method provides an efficient way to solve MTL problems with different trade-off preferences by a single model. See the below responses to AnonReviewer4 for the scalability of our method.
> >
> > 5. There is a concurrent ICLR submission that shares similar idea with our work (https://openreview.net/forum?id=NjF772F4ZZR). They also demonstrate the benefit of reformulating MTL as a solving multiobjective optimization problem, and propose to use hypernetwork to learn the entire trade-off curve. We also find some current works on learning the trade-off curve on image generation [12] and multiobjective reinforcement learning [13,14]. These works show that it is promising to learn the entire trade-off curve for different real-world applications. We will add discussions with these related work in our manuscript.
> >
> > We hope these responses can address your concerns. Please let us know if you have any questions and further concerns. We look forward to additional discussion.
> >
> > [1] Fliege \& Svaiter. Steepest descent methods for multicriteria optimization. Mathematical Methods of Operations Research, 51(3):479–494, 2000.
> >
> > [2] Jean-Antoine Desideri. Mutiple-gradient descent algorithm for multiobjective optimization. In European Congress on Computational Methods in Applied Sciences and Engineering 2012.
> >
> > [3] I. Marusciac. On Fritz-John-type optimality criterion in multi-objective optimization, Mathematica 1982.
> >
> > [4] Dauphin et al. Identifying and attacking the saddle point problem in high-dimensional non-convex optimization. NeurIPS 2014.
> >
> > [5] Jin et al. How to escape saddle points efficiently. ICML 2017.
> >
> > [6] Du et al. Gradient descent finds global minima of deep neural networks. ICML 2019.
> >
> > [7] Kaisa Miettinen. Nonlinear multiobjective optimization 2012.
> >
> > [8] Sener \& Koltun. Multi-task learning as multi-objective optimization. NeurIPS 2018.
> >
> > [9] Lin et al. Pareto multi-task learning. NeurIPS 2019.
> >
> > [10] Ma et al. Efficient continuous Pareto exploration in multi-task learning. ICML 2020.
> >
> > [11] Mahapatra \& Rajan. Multi-task learning with user preferences: Gradient descent with controlled ascent in pareto optimization. ICML 2020.
> >
> > [12] Dosovitskiy \& Djolonga. You Only Train Once: Loss-Conditional Training of Deep Networks. ICLR 2020.
> >
> > [13] Yang et al. A Generalized Algorithm for Multi-Objective Reinforcement Learning and Policy Adaptation. NeurIPS 2019.
> >
> > [14] Parisi et al. Multi-objective Reinforcement Learning through Continuous Pareto Manifold Approximation. JAIR 2016.

---

> ### Author Response · Authors · 2020-11-23
> **Updated Response to R3**
>
> We have updated a revised version of our manuscript where the summary can be found in the above post. Here we want to provide you a point-to-point summary related to your concerns:
>
> 1. **Clarification**: We have revised the whole manuscript to make it clear that the solutions our model can generate are the Pareto stationary solutions. Thank you for pointing this out to let us have a more precise statement on our work.
>
>
> 2. **Novelty and Contribution**: We want to highlight that our contribution is to use **one single model to learn the entire trade-off curve (it might contain infinite Pareto stationary solutions) for an MTL problem**. In contrast, the recent work can only find one single or a set of finite Pareto stationary solutions for an MTL problem [8-11] (and hence they need to train and store multiple large DNN models). A concurrent submission to ICLR also proposes a very similar idea with us (https://openreview.net/forum?id=NjF772F4ZZR) as a novel contribution.
>
>
> 3. **More on Contribution**: Even though our proposed model can learn the trade-off curve, it only has a comparable size with a single MTL model. The proposed model can **scale well for large scale problems and supports real-time trade-off preference adjustment for inference**. It provides a novel and efficient way to solve MTL problems, and could be a valuable contribution to the MTL community. We will also open source our code to let other researchers easily reproduce our results (see supp.zip for code examples).
>
> We hope this revision and responses can address all your concerns. A detailed initial response can be found below. Please let us know if you have further concerns and questions on our work. Thank you again for reviewing our paper.

---

### Author Response · Authors · 2020-11-12
**Initial Response**

We want to thank all reviewers for their efforts and time in reviewing our paper. Here we provide the initial responses to all reviewers' comments, point by point to each reviewer. We are currently working hard on revising this paper, and conducting extra experiments required by the reviewers, and will update the manuscript accordingly. Please let us know if any of our responses are not clear, or you have further concerns and questions.

---

### Author Response · Authors · 2020-11-18
**Revised Manuscript V2 Uploaded**

Dear Reviewers,

We have uploaded the revised manuscript V2 of our work. The major revisions include:

1. Clarification: We have made it clear in the whole paper that the solutions our method can generate are the Pareto stationary solutions.

2. Contribution: We have emphasized that our contribution is to learn the entire trade-off curve  for an MTL problem via a single model.  which supports flexible trade-off adjustment for MTL problems. It is a novel and valuable contribution to the MTL community.

3. Model Details: We have now added more details and discussions for our proposed model in Section 5. The proposed model has a comparable number of parameters with a traditional MTL network, and can scale well for large models and support real-time preference adjustment.

4. Experiments: We have provided the size and inference latency for all models we used, and have added more analysis on the experimental results. We have also added a new experiment with pretrained ResNet-50 backbone, and provided more experiment details in the appendix.

All major modifications are highlighted in **blue.**

We sincerely hope this revision with the previous responses can address the current concerns. Further revision (most for the appendix) will be updated later. Please let us know if you have any further concerns and questions.

Paper1412 Authors

---

### Author Response · Authors · 2020-11-23
**Updated Response to All Reviewers**

Dear Reviewers,

We have uploaded the revised Manuscript V2 of our work. We sincerely thanks all reviewers for their comments and suggestions, and have carefully addressed each of them in the revised manuscript.

The major modifications are highlighted in blue, and a summary can be found in the post below. We also provide a point-to-point response summary to each reviewer.

We believe our paper is significantly improved by considering all reviewers' comments, and we sincerely hope the revised manuscript, together with our response, can resolve the concerns raised by the reviewers.

Paper1412 Authors

---

### Decision · Program_Chairs · 2021-01-07
**Final Decision**

**Decision:**

Reject

**Comment:**

The paper first aims to propose a new controllable Pareto multi-task learning framework to find pareto-optimal solutions. But after the revision according the comments, the paper claims to find finite Pareto stationary solutions. But the paper still can not prove their proposed method can find the Pareto stationary solutions. Even if they can find the  Pareto stationary solutions, they can not guarantee find the pareto front which is conflict with the experiments and claims. There are major flaws in the paper.